# Exosome-Mediated Antigen Delivery: Unveiling Novel Strategies in Viral Infection Control and Vaccine Design

**DOI:** 10.3390/vaccines12030280

**Published:** 2024-03-07

**Authors:** Daed El Safadi, Alexandre Mokhtari, Morgane Krejbich, Alisé Lagrave, Ugo Hirigoyen, Grégorie Lebeau, Wildriss Viranaicken, Pascale Krejbich-Trotot

**Affiliations:** 1Unité Mixte Processus Infectieux en Milieu Insulaire Tropical (PIMIT), Université de la Réunion, INSERM U1187, CNRS UMR 9192, IRD, Plateforme Technologique CYROI, 94791 Sainte Clotilde, La Réunion, France; daed.el-safadi@univ-reunion.fr (D.E.S.); alexandre.mokhtari@universite-paris-saclay.fr (A.M.); gregorie.lebeau@univ-reunion.fr (G.L.); wildriss.viranaicken@univ-reunion.fr (W.V.); 2Centre de Recherche en Cancérologie et Immunologie Intégrée Nantes-Angers, CRCI2NA, INSERM U1307, CNRS UMR 6075, Université de Nantes, Université d’Angers, 8 Quai Moncousu, P.O. Box 70721, Cedex 1, 44007 Nantes, France; morgane.krejbich@etu.univ-nantes.fr (M.K.); ugo.hirigoyen@inserm.fr (U.H.); 3National Reference Center for Arboviruses, Institut Pasteur de la Guyane, Cayenne 97300, French Guiana; 4Unité Mixte Diabète Athérothrombose Réunion Océan Indien (DéTROI), Université de La Réunion, INSERM U1188, Campus Santé de Terre Sainte, 97410 Saint-Pierre, La Réunion, France

**Keywords:** exosomes, extracellular vesicles, cargoes, pathological biomarkers, infection, viruses, flaviviruses, antigen presentation, exosome engineering, vaccine design

## Abstract

Exosomes are small subtypes of extracellular vesicles (EVs) naturally released by different types of cells into their environment. Their physiological roles appear to be multiple, yet many aspects of their biological activities remain to be understood. These vesicles can transport and deliver a variety of cargoes and may serve as unconventional secretory vesicles. Thus, they play a crucial role as important vectors for intercellular communication and the maintenance of homeostasis. Exosome production and content can vary under several stresses or modifications in the cell microenvironment, influencing cellular responses and stimulating immunity. During infectious processes, exosomes are described as double-edged swords, displaying both beneficial and detrimental effects. Owing to their tractability, the analysis of EVs from multiple biofluids has become a booming tool for monitoring various pathologies, from infectious to cancerous origins. In this review, we present an overview of exosome features and discuss their particular and ambiguous functions in infectious contexts. We then focus on their properties as diagnostic or therapeutic tools. In this regard, we explore the capacity of exosomes to vectorize immunogenic viral antigens and their function in mounting adaptive immune responses. As exosomes provide interesting platforms for antigen presentation, we further review the available data on exosome engineering, which enables peptides of interest to be exposed at their surface. In the light of all these data, exosomes are emerging as promising avenues for vaccine strategies.

## 1. Introduction

Cellular communication is an essential aspect of multicellular organism physiology. It relies on signals emitted by cells, capable of circulating to reach target cells specifically equipped to receive the message. Signal production and reception require complex cellular and molecular networks. By implementing multiple biological programs, these networks enable a wide range of coordinated cellular responses. Responses vary from phenotypic and metabolic changes to the control of movement and cell migration, the modification of gene expression, and the determination of cell survival, division, differentiation, or death. Therefore, communication systems condition the constant adaptability of cells to their environment. The underlying mechanisms have been extensively explored for decades, notably in the field of immune cell interactions, where coordinated immune responses are required to eliminate pathogens or tumor cells. Intercellular signaling between immune cells was initially thought to occur either through cell–cell contact or soluble factors, such as cytokines, until the late 1990s when it was identified that extracellular vesicles (EVs) were involved in dendritic and T cell crosstalk [1,2] and constituted antigen-presenting devices [3,4]. This discovery marked a turning point in the understanding of immune cells crosstalk. It demonstrated that EVs produced by immune cells could encapsulate cytokines, circulate, contact, or deliver their content to recipient cells, thus priming or immunomodulating other immune cells [5]. Since then, EVs have emerged as key players in intercellular communication, and their nature and biological functions have been increasingly documented. Owing to their membrane-bound nature, EVs provide a unique facility for transporting a wide variety of signaling molecules and driving their precise targeting to recipient cells.

In recent years, the study of EVs has been greatly facilitated by the development of new and improved methods. These methods enable their detection and purification from a variety of biological fluids, as well as their imaging, size measurement, and content analysis through lipidomics, proteomics, and genomics studies [6,7]. It is worth highlighting that major technological advances now enable specific analysis at the level of single vesicles. These significant developments, regularly reviewed, make it possible to collect extremely valuable precision biometric and molecular information on EVs [8]. Research in the field has revealed that EVs and their cargoes are highly diverse in nature. From this observation ensued a need for a classification, which first put emphasis on the membrane origin: vesicles derived from the plasma membrane are considered as ‘ectosomes’, whereas vesicles that originate from an endosomal biogenesis pathway are named ‘exosomes’ [9,10]. Moreover, their biogenesis, production, and content differ according to a plethora of parameters, such as cell type, metabolic activity, and physiological and pathological conditions. This plasticity complexifies the understanding of the biological activities of EVs. At the same time, their morphological and molecular signatures make them powerful biomarkers and indicators of the progression of numerous diseases.

In this review, we have chosen to highlight exosomes as important actors in the context of viral infections and as potential biological tools that could pave the way for new strategies to combat these infections. Our emphasis on exosomes among all EVs is grounded in the notable ability of viruses to corrupt the exosome biogenesis pathway, and the extensive data related to the transport of viral components by exosomes. Their use as biomarkers in the context of viral infections is also an area of study that is becoming well documented. In addition, their ability to present antigens and the development of engineering methods to tailor their content make them interesting therapeutic tools, and especially promising immunogenic tools for vaccine development.

In light of the latest data in the literature, our aim was firstly to provide insight into the known features of exosomes. We sought to understand how their biogenesis, specific loading, and interaction with targeted cells can lead to specific physiological responses. We then provided an overview of what is known about exosomes in the biology of viral infection, notably that of flaviviruses. In doing so, we sought to unveil the dual roles of exosomes, which are powerful players in the immune response, permitting the resolution of infections and viral clearance. Conversely, exosomes are often hijacked to promote virus dissemination or the vectorization of viral factors, contributing to infectious pathophysiology. We then focused on the potential of exosomes to be reliable indicators of health and disease, turning them into valuable biomarkers. After a brief overview of the potential of exosomes as diagnostic tools, we also addressed their interest as therapeutic tools. This involves engineering methods aimed at controlling exosome content to deliver drugs. In line with the abundant literature examining the function of exosomes in mounting adaptive immune responses during viral infections, our main goal was ultimately to explore the capacity of exosomes to vectorize immunogenic viral antigens. As exosomes provide interesting platforms for antigen presentation, we reviewed the available data on exosome engineering, which allows peptides of interest to be addressed at their surface. In the light of all these data, we propose exosomes as emerging and promising avenues for vaccine strategies. Figure 1 provides an overall diagram outlining the different goals of our review, with a particular focus on exosomes in the context of viral infections. Additionally, it highlights their role as manipulable tools for infection control and vaccine design.

## 2. Exosomes Are Small, Specialized Extracellular Vesicles (EVs)

Extracellular vesicles (EVs) refer to all lipid vesicles released by cells. Despite an ever-changing nomenclature, EVs are distinguished by their biogenesis pathways, morphology, molecular composition, and size [11,12]. According to a substantial body of literature, the size of EVs serves as a reliable indicator in determining their nature and origin. This led to the identification of three subcategories of EVs: first, blebs and apoptotic bodies, with sizes reaching up to 5 μm; then, microvesicles, also called ectosomes, ranging between 100 and 500 nm, and sometimes even up to 1000 nm; lastly, exosomes, which exhibit diameters varying between 30 and 150 nm [13,14]. The first two subcategories are produced by shedding from the cell surface, with blebs being generated in the late stages of apoptosis. Their membranes derive from the plasma membrane of the donor cell. In contrast, exosomes proceed from biogenesis pathways and derive from endosomal membranes. They are released through exocytosis and follow cell secretion pathways.

EVs are constitutively produced by a wide variety of cell types. The capacity of cells to release vesicles appears to be a highly conserved mechanism throughout evolution, and has even been observed in bacterial communities [15,16]. As a consequence, large quantities of EVs can be detected in extracellular compartments. In humans, exosomes can be found in almost all body fluids, including blood, cerebrospinal fluid, saliva, breast milk, and urine [13,17,18,19]. When first discovered in the late 1980s, exosomes were initially considered as cellular waste. Nevertheless, it has been shown that EVs play a crucial role in intercellular communication, both in physiological and pathological conditions, and maintain homeostasis at cellular and systemic levels [20,21]. This role is fulfilled by their ability to encapsulate and transport diverse yet specific cargoes of complex biochemical composition: as it is, proteins, lipids, secondary metabolites, and nucleic acids have all been found within exosomes. This characteristic positions them as true unconventional secretory systems [22].

The action of EVs has been described to either occur in the vicinity of their production site, thus promoting autocrine or paracrine signaling, or at a distance from their secretion site, serving as stable conveyors of signaling molecules with endocrine action [23]. Exosome interaction at the surface of recipient cells, and their potential uptake, may elicit pleiotropic responses and induce phenotypic changes [24]. The in vitro biological effects of exosome exposure have been extensively documented. However, the exact physiological or pathological roles of exosomes in vivo are yet to be fully understood, offering a significant area for further research. Their production and behavior in the biology of viral infections deserve particular attention, as detailed below. Indeed, their importance as molecular signatures and prognostic biomarkers in the evolution of infectious diseases and several other pathologies will be explored in subsequent paragraphs. Due to the close proximity between the intracellular sites of exosome production and those of viruses, a strong inter-relationship should necessarily be established in infected cells. Therefore, a comprehensive understanding of exosome biogenesis is particularly relevant for studying their role during viral infections.

### 2.1. Exosome Biogenesis

Exosomes are small EVs that have their own biogenesis pathway within cells, prior to their release into the extracellular environment. Krylova and Feng recently reviewed the extensive literature dealing with their biogenesis mechanisms [25]. Exosome formation occurs in cellular compartments called multivesicular bodies (MVBs), which proceed from the maturation of endosomal compartments [26]. This process is illustrated in Figure 2A. Endosomes are intracellular sites of convergence for vesicular trafficking originating mainly from the cell surface [27]. A broad range of substances can travel through the endomembrane compartments [28,29]. Following their internalization by receptor-dependent endocytic pathways, captured molecules reach early endosomes [30]. The endosomal compartment includes recycling components, which undergo centrifugal vesicular trafficking back to the plasma membrane. Alternatively, early endosomal components undergo maturation in late endosomes with more acidic content, closely related to autophagosomes. The remodeling of late endosomes into MVBs occurs after the invagination of their membrane. This results in the formation of a multitude of small vesicles called intraluminal vesicles (ILVs) [31,32]. In the course of this remodeling process, ILVs can be loaded with exceptionally diverse molecules ranging from cytosolic proteins to metabolites, lipids, and even nucleic acids. An exocytosis mechanism allows MVBs to fuse with the cell membrane, releasing ILVs into the extracellular compartment where they become exosomes. Otherwise, MVBs are routed to the lysosomes where their content is degraded, including membrane receptors [33,34]. The ILV genesis process relies on an intricate protein machinery called the Endosomal Sorting Complex Required for Transport (ESCRT) which reshapes the endosomal membrane topology [35,36]. During MVB formation, the ESCRT allows the intraluminal budding of vesicles and the sorting of embedded cargo proteins. The ESCRT comprises around twenty proteins, including ALG-2-interacting protein X (ALIX), tumor susceptibility gene 101 (TSG101), and charged multivesicular body protein 4a (CHMP4). The molecular actors of ESCRT are organized into four cooperative complexes called ESCRT-0, -I, -II, and -III, which are conserved from yeast to mammals [37]. The ubiquitination of proteins located on the endosomal membrane drives the inward budding. ILVs containing ubiquitinated proteins are destined to be degraded into the lysosome. The de-ubiquitination of these proteins by the de-ubiquitylating enzymes (DUBs) will rescue the ILVs from degradation. Numerous studies have also shown that phosphoinositides, ceramides, and the lipid composition of the endosomal membrane as a whole are key factors in the biogenesis of ILVs [38,39]. It is also worth mentioning that ESCRT-independent mechanisms have been delineated as alternative systems for the formation of ILVs, and that vesicle release by cells could also occur from lysosomes [40,41,42]. In addition, a process involving tetraspanin enrichment appears to be important for membrane remodeling and ILVs formation [43,44]. Tetraspanins are a family of transmembrane proteins including CD9, CD63, and CD81, which are abundantly found in MVBs and ILV membranes [45,46]. They are therefore frequently used as markers for identifying EVs of exosomal origin [47]. However, it is important to emphasize that the presence of CD9 and CD81 has also been observed on the plasma membrane. Hence, CD63 remains the most specific tetraspanin marker for exosomes [48]. Tetraspanins are known to participate in the cargo sorting process that occurs during ILV formation and could also play a role in determining the specificity of exosomes for particular recipient cells [44,49]. The selective process of cargo sorting and the specific components required for the biogenesis of exosomes have a tremendous impact on their final composition.

The final stage in the cellular production of exosomes is their release into the extracellular environment. Exosome secretion by cells appears to be a continuous mechanism but with quantitative and qualitative controls. It is known that specific interactions between Rab proteins and SNARE (soluble NSF attachment protein receptor) complexes control the intracellular trafficking of MVBs and determine their subsequent destination [50,51,52,53]. An increase in the quantities of circulating exosomes has been particularly noticed in many pathological situations. The increased production and secretion of exosomes with modified content is indeed part of cellular responses to various stresses. This confirms the role of exosomes as vectors of mediators, assisting cell to cell communication in these stressful situations. The internal and environmental factors regulating MVB exocytosis are not yet fully understood. According to Xu et al., these include, but are not limited to, intracellular Ca^2+^ concentrations, MVB acidification processes, nutrient and oxygen availability, and oxidative metabolism. Additionally, various signaling pathways, including those of growth factors, contribute to this complex regulatory network [54].

In conclusion, the biogenesis and secretion of exosomes emerge as pivotal biological programs orchestrated by our cells to navigate through diverse physiological situations, environmental changes, and stresses. This program aims to emit vesicular communication devices that will complement the arsenal of soluble mediators. These vectored messages will in turn inform recipient cells, which will respond by adapting their biological activities. Various stimuli lead to the subsequent reprogramming of biogenesis mechanisms, resulting in the repackaging of exosomal cargo. Therefore, the molecular composition will determine the nature of the message and its targets.

### 2.2. Exosomes Composition

A non-exhaustive schematic representation of a canonical exosome, with some of its typical compounds and cargoes, is illustrated in Figure 2B. During their formation, ILVs are loaded with both membrane and soluble cytosolic proteins. Some components, mostly involved in their biosynthesis process, are common denominators further shared by all released exosomes. Tetraspanins, syntenin-1, TSG101, and ALIX are considered molecular signatures of exosomes and, in this regard, could serve as reliable markers, differentiating exosomes from other EVs [55]. Tetraspanins (i.e., CD9, CD63, and CD81), which are strongly represented in the membrane, play an important role in cell contact, uptake, and membrane fusion events [56]. Embedded within the membrane, various adhesion molecules (e.g., integrins, ICAM-1) have been identified [57]. Adhesion molecules could allow the docking of exosomes at the surface of recipient cells, prior to their internalization. The diverse nature of adhesion molecules, as well as their relative abundance at the exosome surface, are likely influencing their selectivity for recipient cells. Thus, depending on their relative abundance at the exosome surface, these membrane adhesion factors may reflect pathological situations. In cancer, for example, integrin display patterns have been closely associated with the organotropism of tumor exosomes, whose uptake contributes to tumorigenesis [58,59]. Additionally, major histocompatibility complex (MHC) type I molecules can be found in the membrane of exosomes, albeit in variable quantities according to cell type and local microenvironment. MHC class II and co-stimulatory molecules are more restricted to exosomes secreted by professional antigen-presenting cells (APCs). However, both MHC-I and MHC-II are required for mature exosomes to prime naive T cells [60]. Interestingly, the presence of functional MHC–antigenic peptide complexes could confer exosomes with a prominent ability to initiate adaptive immune responses [4,21]. Regarding the protein content of exosomes, various chaperones are loaded into exosomes and use this alternative secretion pathway to join in extracellular networks. These include members of the heat-shock proteins (HSPs) and J-domain proteins [61]. In addition, positive correlations have been described between HSP70 levels in exosomes and cancer malignancy [62]. Of note, the presence of traditionally abundant proteins in exosomes, such as cytoskeletal proteins, GAPDH, and even HSPs, has become a topic of controversy. These uncertainties regarding exosome content arise from advances in techniques for separating small vesicles and the growing availability of powerful analytical methods, such as mass spectrometry, for studying their proteomes [63]. Other exosomal components are more specific and may be uniquely related to cell types and their physiological state [1,64,65]. Thus, the content of exosomes may reflect the molecular composition of the producing cell, and indicate the specific biological activities elicited in the recipient cells [66,67]. Importantly, cells are able to adjust their exosome production both qualitatively and quantitatively, further supporting the implication of EVs in intercellular communication. This also suggests that the content of ILVs is not the mere result of passive diffusion, the loading process being actively orchestrated by cells. As mentioned in the previous paragraph, the control of exosome release and packaging depends on a variety of stimuli, allowing the secretion of exosomes carrying fine-tuned messages [43].

**Figure 2 vaccines-12-00280-f002:**
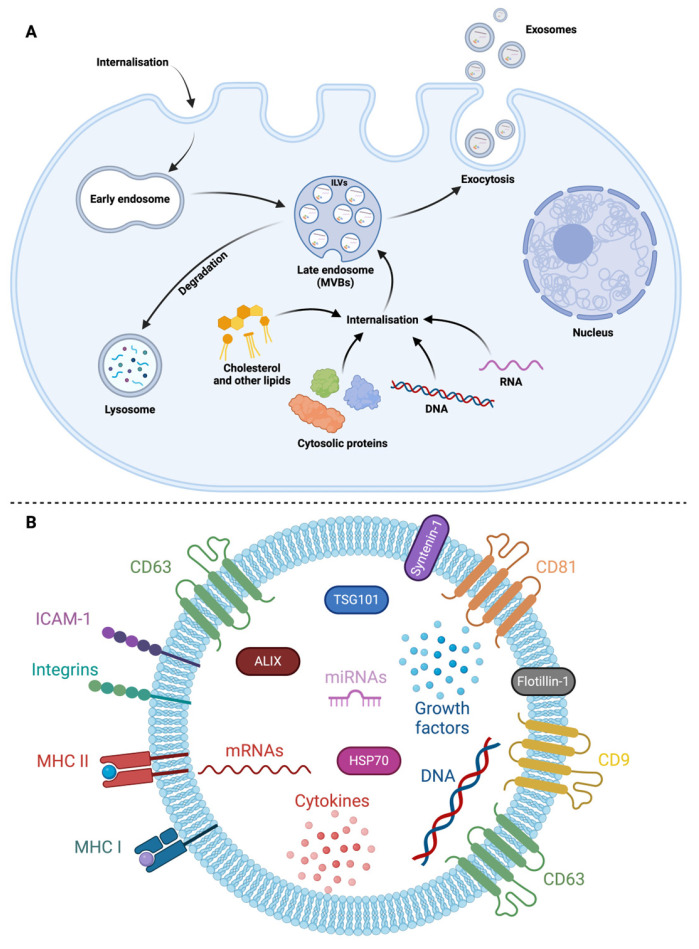
Insights into exosome biogenesis and composition. (**A**). Exosomes biogenesis. The early endosomes produced by the endocytic pathways can mature into multivesicular bodies (MVBs). During their formation, the intraluminal vesicles (ILVs) are loaded with cytosolic material like proteins, lipids, and nucleic acids. MVBs have two possible fates. They are either directed to lysosomes, enabling the degradation of ILV constituents, in particular membrane components, or released by exocytosis, enabling the externalization of ILVs, then called exosomes. (**B**). Schematic representation of an exosome and its standard cargo. Exosomes are delimited by a phospholipid bilayer membrane containing proteins from the tetraspanin family, such as CD9, CD63, or CD81. The intercellular adhesion molecule 1 (ICAM-1) and major histocompatibility complex MHC I and/or II are detected in variable proportions in the membrane, depending on the emitting cells, and contribute to the biological or pathological functions of exosomes. The ALIX protein is identified within these vesicles, serving as one of the primary intravesicular markers. Additional intra-exosomal markers exist, such as the tumor susceptibility gene 101 (TSG101), which plays a critical role in exosome biogenesis and secretion. A variety of chaperones, including members of the heat shock protein 70 (HSP70) family, could use this unconventional secretion pathway. In addition to proteins, these vesicles can carry nucleic acids, including DNA and RNA such as miRNA. Adapted from Kowal et al. [57]. Created with Biorender.com.

In response to specific contexts, cytokines and various mediators seem to be able to use the exosomal vectorization pathway. This has been documented for interleukin-1β (IL-1β), which can be loaded and transported via exosomes in inflammatory and pyroptotic contexts [68,69]. IL-6, IL-10, and TNF-α could also be transported in this way, giving these mediators a longer lifespan in body fluids and reinforcing exosomes as suitable markers for prognosis and diagnosis [70]. The broad diversity of exosomal content also suggests a trafficking between cellular compartments that remains poorly understood. For example, MVBs and autophagosomes have been shown to interact, allowing cargo exchange. This proximity may also favor the release of exosomes [71].

Besides proteins, exosomes contain lipids which shape their membrane and contribute to their biogenesis pathway [72]. Sphingomyelin, phosphatidylserine, glycosphingolipids, and regulated concentrations of cholesterol are the hallmarks of ILVs and exosome membranes. Thus, cholesterol-enriched MVBs appear more prone to lysosomal digestion, while lower cholesterol levels are more associated with the exocytosis pathway [73].

Exosomes also carry nucleic acids like short DNA molecules, small RNAs, and especially miRNAs [74,75]. Deep sequencing RNA techniques confirmed that many types of non-coding RNAs were loaded into ILVs. Their composition depended on cell types and cellular contexts. Several heterogeneous nuclear ribonucleoproteins (HNRNPs) and RNA-binding proteins have been shown to promote RNA sorting and loading in ILVs. Additionally, they could play a role in RNA function in target cells. These mechanisms were extensively reviewed by Corrado et al. [76].

Remarkably, and in the light of numerous data in the literature, it is clear that exosome composition in terms of miRNAs undergoes significant changes under different physiological and pathological conditions. For instance, studies on the effects of exposure to a set of environmental factors, known as the ‘exposome’, on carcinogenesis processes have demonstrated an unexpected role for exosomes. Data recently reviewed by Kalita-de Croft et al. report that exosomes produced after the exposure of cells to various chemicals, including arsenite and other substances present in cigarette smoke, exhibited an altered miRNA content. This finding highlights exosomes as sensors of these exposures and as biological players ultimately participating in pathophysiological processes [77].

This variability is well regarded for its prognostic and diagnostic properties. Consequently, the quantity and content of exosomes derived from patients plasma emerge as a valuable tool for monitoring the progression of diverse pathologies [78], from autoimmune diseases such as rheumatoid arthritis [79,80] to cancers [81]. These aspects will be elaborated further below. The observation of changes in exosome composition under pathological conditions also suggests that the molecular variability, as well as the way exosomes interact with their target cells, are at the origin of a broad range of biological responses in the recipient cells.

### 2.3. Exosome Biological Activities according to Their Mode of Interaction with Recipient Cells

The enrichment of exosomes with various components underscores their intricate nature and the potential for functional diversity. There is extensive literature on the messenger functions of exosomes, during immune responses or pathophysiological situations like tumor development or host–pathogen interactions [82]. Recipient cells may be near or distant from the exosome-shedding cells and are reached because EVs can circulate in a wide range of biological fluids. It can naturally be inferred that the vesicle-mediated transport of biological mediators may protect them from degradation; however, this feature has not been fully verified in vivo. While EVs can remain stable in serum for days, their half-life in circulation is notably brief. Studies in mice indicate that exogenous EVs exhibit a very short half-life (2–10 min) in the blood circulation [83]. This brief circulation time could, however, be attributed to a rapid uptake of circulating EVs by scavenger cells like monocytes and endothelial cells. Indeed, several studies have shown that small EVs have no difficulty in crossing endothelial and blood–brain barriers, sometimes by transcytosis, and thus diffuse widely into tissues [84,85,86].

Exosomes appear to be able to interact with their target cells in various ways [53,64]. Firstly, they can act through direct contact with cell surfaces. This is the case for exosomes that carry MHC–peptide complexes and can induce the activation of TCR receptors on T lymphocytes [60]. Another example is given by exosomes produced by melanoma cells that are loaded with FasL ligand during their biogenesis. They can induce apoptosis in recipient lymphocytes by mere contact [87]. Although a simple contact between exosomes and their recipient cells is sometimes sufficient to trigger signaling pathways, effective phenotypic changes are more likely to be achieved through the transfer of exosome content into recipient cells. The ability of exosomes to deliver their cargoes has been a longstanding hypothesis, and it has been demonstrated only recently. The release of intra-exosomal molecules into the cytoplasm is achieved by membrane fusion, either directly with the plasma membrane or after endocytosis and transit through the endosome where a back-fusion process takes place. A functional assay has been developed to verify that membrane fusion, exposing the luminal face of the exosome to the cytoplasm, occurs following exosome uptake. The assay is performed using exosomes engineered to vectorize a transcription factor which enables the expression of a reporter gene upon its intracellular delivery [88]. Thanks to this demonstration, the ability of exosomes to deliver their protein content has been assessed and is regularly reviewed [89]. Similarly, it has been shown that small RNAs and miRNAs carried by exosomes can be transferred into recipient cells. Once inside, they retain full functionality and can modify the behavior of the recipient cells through gene expression modulation [90]. The growing body of data on the ability of exosomes to specifically vectorize and deliver various types of molecules to cells has clearly opened up tremendous new avenues for their use as nano-therapeutic tools.

Recipient cells can internalize exosomes through an uptake process that can be visualized through several imaging techniques [91]. The mechanisms by which recipient cells take up exosomes have been the subject of extensive research for over 20 years [92,93]. Although macropinocytosis may be a potential entry mechanism, the capture of exosomes appears to be quite specific. However, the recognition process by which exosomes are identified by their target cells remains poorly understood, and receptors specialized in exosome uptake have not been formally identified yet [94]. It has been proposed that T-cell immunoglobulin- and mucin-domain-containing molecule 4 (TIM4) could bind to exosomes through the phosphatidylserine exposed at their surface [95,96]. Other studies have suggested that exosome endocytosis may result from their interaction with heparan-sulfate proteoglycans [97], or syndecan 4 [98]. Once internalized, exosomes have several possible fates. They may follow a recycling path, and before being returned to the extracellular environment, acquire new cargoes. They may undergo lysosomal digestion, or release their cargo into a recipient cell cytoplasm through a pH-dependent fusion process with endosomal membranes [99]. The factors and mechanisms that govern the fate of exosomes after the internalization process remain to be fully elucidated.

Once they have reached their target cells, exosomes perform their messenger functions. As in intercellular communication based on soluble signal molecules and membrane signal receptors, these responses are behavioral or phenotypic. The biological effect of exosomes has been particularly well studied in pathological contexts, in order to elucidate their positive or negative role in disease progression, with a view to identify new therapeutic players or targets. Their role in cancerous processes has notably received particular attention, enabling the scientific community to unravel the mechanisms of exosome capture and action. At the same time, the communication system driven by EVs in cancer revealed an extreme complexity of both pro- and anti-tumor effects [100,101,102]. For example, EVs derived from malignant cells, which differ in content from exosomes produced by healthy cells, have been shown to play a key role in liver pre-metastatic niche initiation during pancreatic cancer [58,59]. Moreover, exosome transfer from highly metastatic to poorly metastatic tumor models has been shown to increase the metastatic behavior of the latter model [103]. It remains to be elucidated how exosomes shape the microenvironment of those pre-metastatic niches. However, the selective uptake of exosomes by Kupffer cells in the liver was proposed to induce the activation of fibrotic pathways and a pro-inflammatory environment that supports metastasis formation [104]. Mechanisms such as vascular permeability impairment or local immunosuppression may be involved. Indeed, exosomal hepatocyte growth factor (HGF) receptor was shown to skew hematopoietic stem cells towards pro-vasculogenic phenotypes. Additionally, C-C motif chemokine ligand 2 (CCL2) facilitates the uptake of tumor exosomes by immune cells, which ultimately leads to the enrichment of myeloid suppressive cells at future metastatic sites [105,106]. The communication system woven through EVs appears to be particularly prevalent during carcinogenesis, suggesting that cancer cells have evolved exosome-based strategies to better adapt to their microenvironment. For example, it has been shown that the exosomal transfer of oncomir-like miRNAs by cancer cells induces the reprogramming of recipient cells and promotes prometastatic behavior [107]. Other exosomal miRNAs are suspected to be involved in immune escape by promoting the anergy (activation-induced non-responsiveness) of CD8+ T lymphocytes [108]. A decrease in cytokine production and granzyme B secretion was also observed after the internalization of tumor exosomes by lymphocytes. Moreover, exosome-mediated signaling may not only protect cancer cells from the host immune system, but also contribute to their resistance against anti-cancer therapies. This mechanism has been demonstrated in diverse cancer types, encompassing hematological malignancies, as well as colorectal, breast, prostate, liver, and lung cancers [109]. In their study on pancreatic cancer, Patel et al. demonstrated that gemcitabine-treated pancreatic cancer cell lines could produce exosomes that increased the IC50 (half maximal inhibitory concentration) of non-treated cancer cells [110]. Strikingly, cancer cells that had never been directly exposed to a drug could still display a significant resistance to it. Thus, in the context of cancer, exosomes could play a determinant role in the migration and proliferation of tumor cells, but also in the establishment of a complex communication network with their neighboring cells. This could actively contribute to angiogenesis, remodeling the tumor immune microenvironment by disrupting the conventional cross-talk between immune cells, thereby promoting the formation of tumor niches [111,112]. The dual function of exosomes in the tumor context, with their strong capacity to ensure communication but with adverse pro-tumor effects, has led to the development of two main types of exosome-based anti-cancer strategies. One is based on the inhibition of EV trafficking to improve new therapeutic combinations [113]. Another is based on the engineering of exosomes to use them as nano-vectors for therapies [114].

Considering the amount of research carried out on the cellular responses and phenotypic changes induced in cells exposed to exosomes, it is very challenging to give an exhaustive overview. Among other examples of phenotypic changes induced, exosomes were shown to contribute to the polarization of macrophages [115]. Exosomes have also been described as players in epithelial–mesenchymal transition (EMT) in a variety of pathological conditions, including fibrosis and cancer [116,117]. However, the miRNA content of exosomes, depending on the emitting cells, would mediate potentially antagonistic effects on EMT, revealing both the promotion and inhibition roles [118,119].

In conclusion, the vast amount of data on the biological effects of exosomes reveals the multifaceted nature of exosome-mediated cellular communication. The substantial development of research into their activities in the field of cancer confirms their key role in the progression of these diseases. Meanwhile, this intensive research has also revealed the difficulty of exploring the fate of exosomes in vivo, leading to potential controversies and limitations. Despite the often ambiguous and challenging-to-delineate activities of exosomes, harnessing their knowledge could help find sustainable applications, including innovative diagnostic approaches and targeted therapeutic interventions. This paves the way for further exploration of personalized medicine and precision therapeutics.

Although the interest in exosomes in the field of infectious diseases is more recent, there is a constantly growing number of studies on the role they could play in the resolution of viral infections, in the clearance or persistence of viruses, and in the evolution of infectious pathophysiology. In addition, specific molecular signatures linked to the vectorization capacities of viral factors reveal that exosomes could serve as biomarkers for monitoring infectious diseases. A detailed overview of the data in this area deserves our attention.

## 3. Exosomes and Biology of Infection

The particular biological context arising from the infection of tissues has prompted scientists to investigate whether exosomes could exhibit a modified profile. Numerous studies have shown that exosome biogenesis in infected cells was altered, leading to increases in their production and modifications in their content. Such results were obtained for in vitro infections with several phylogenetically unrelated families of both RNA and DNA viruses, and confirmed by analyses of biological samples, e.g., patient plasma [120,121,122,123,124,125]. In particular, an increase in exosome quantity has been revealed in the case of infection with flaviviruses, notably Dengue (DENV) and Zika (ZIKV) viruses. This was observed in vitro during the first 24 h of infection and was accompanied by a change in EV size [126]. The ability of exosomes to move bidirectionally from the plasma membrane, and across the endosomal pathway, has aroused suspicions about their possible involvement in the dissemination and pathogenesis of intracellular pathogens (viruses, parasites, and intracellular bacteria). However, it is increasingly documented that exosomes play an active role in mounting the immune response, which commonly provides beneficial effects to the host in the course of an infection. In this paragraph, we will attempt to review the literature and expose the striking ambivalence of exosomes during viral infections, the main aspects of which are summarized in a schematic representation (Figure 3).

### 3.1. Exosomes May Contribute to Viral Dissemination

Many viruses exploit intracellular trafficking mechanisms to complete their multiplication cycle. They initially interact with the cell surface and commonly follow pathways leading to endosomes. The assembly and budding of enveloped viruses share many similarities with the biogenesis of small EVs, as both processes rely on cellular membranes. Consequently, it is unsurprising that some viruses may interfere with the biogenesis of MVBs and the production of exosomes. In particular, the ESCRT system, which is required for the formation of ILVs, has been found to be an essential tool for the production of enveloped viruses. Numerous studies have shown that many viruses hijack the ESCRT system to promote their own budding and, thus, the efficiency of their production and dissemination [127,128]. Flaviviruses are particularly concerned, and numerous studies have shown the dependence of their viral cycle on ESCRT-dependent membrane remodeling [129,130,131].

In line with these findings, it can be hypothesized that MVB formation and the release of exosomes are disrupted, and this defect may favor the virus at the expense of homeostasis control. The hijacking of exosome biogenesis might additionally confer the ability of viruses to target crucial components within exosomes, including their own viral nucleic acids, as well as viral proteins or even complete virions. This was notably shown with the hepatitis B virus (HBV) [132], and since the role of EVs in infections has been studied, examples of viruses using host exosomes as transport systems for their factors have become widespread and are regularly and recently the subject of comprehensive reviews [133,134]. This ability has led to a Trojan Horse hypothesis that exosomes actively contribute to infectious pathogenesis, helping some viruses to evade immune system recognition and sometimes facilitating viral persistence [135]. Hence, the hepatitis C virus (HCV), like other *Flaviviridae* family members (e.g., flaviviruses like DENV, ZIKV, or West Nile virus), enters through endocytosis and releases its genome by a “back fusion” phenomenon which is comparable to the one observed for exosomes [136,137]. But a particularity of HCV is that its genome can be included in ILVs, following the endocytosis of the virus and its transit through endosomes. This viral genetic material is then released into exosomes [137]. Strikingly, these exosomes were shown to act as pseudo-viral particles that can efficiently infect recipient cells [138]. This alternative transport of viral material could even diversify the tropism of viruses, which is thought to be restricted by the display of appropriate receptors on permissive cells. In their recent review of the literature on EVs in Flaviviridae pathogenesis, Latanova et al. cite several studies supporting the ability of EVs to assist neurotropic flaviviruses such as ZIKV and WNV in their neuroinvasive capacities [139]. Similarly, exosomes derived from epithelial cells infected with the Chikungunya virus (CHIKV), a member of the alphavirus genus, have been shown to transport viral RNA and proteins. These exosomes were able to infect naive epithelial cells [140]. This unexpected mode of transmission raises questions about the general ability of viruses to use exosomes as alternative routes of entry into cells [141]. This mimicry with a physiological mode of EV transport and capture could indeed diversify the cellular tropism of viruses and contribute to their escape from the immune system.

Viruses could also use exosomes as a way to transport non-infectious subgenomic RNAs, thus promoting viral replication and dissemination. During human immunodeficiency virus (HIV) infection, HIV-associated RNAs (including TAR, trans-activation response element) have been detected in infected T cell-derived exosomes. These RNAs were able to down-regulate apoptotic signals in recipient cells, thus presenting an improved support of HIV replication [142]. Viral RNAs could also contribute to an accentuation of pro-tumor profiles, especially in Head and Neck Squamous-Cell Carcinoma (HNSCC) cancers. Indeed, it has been shown that infected T cell-derived exosomes carrying TAR promoted the proliferation of cancer cells and induced proto-oncogene expression in recipient cells [143].

In addition to viral RNAs transported in exosomes, virus-derived miRNAs could also exert numerous adverse effects in recipient cells. For example, miRNAs encoded by the Epstein–Barr virus (EBV) and produced by infected cells are able to inhibit genes involved in antiviral activities and inflammation in neighboring non-infected cells through exosomal transfer [144]. During Japanese encephalitis virus (JEV) infection, activated microglial cells release let-7a/b miRNAs through extracellular vesicles [145]. These miRNAs activate caspases in uninfected neuronal cells, potentially contributing to bystander neuronal death. Additionally, in COVID-19 patients, SARS-CoV-2 (Severe Acute Respiratory Syndrome Coronavirus 2) RNA was detected in plasma-derived exosomes, with a distinct proteomic signature. Based on the proteomic analysis, these circulating exosomes could potentially contribute to various processes like coagulation, inflammation, and immuno-modulation during SARS-CoV-2 infection [146].

Viral proteins also use exosomes as effective delivery systems. For example, during HIV infection, the viral membrane protein Gag and the virulence factor Nef have been identified in exosomes derived from infected cells [147,148]. Of note, exosomal Nef was shown to restore the infectivity of Nef-deficient HIV virions in recipient cells. Exosomal Nef also triggers apoptosis in CD4+ T cells and endothelial cells, contributing to the pathophysiology of the infection [149]. On further note, Nef attenuates the effect of RNA interference in recipient cells due to a modification of the miRNA content of exosomes [150]. Other viruses are known to promote the incorporation of their proteins into exosomes, which facilitates their replication or enhances the susceptibility of recipient cells. Such mechanism has notably been reported for the glycoprotein B of cytomegalovirus (CMV) [151], as well as the E2 protein of HCV [138]. Mishra et al. reviewed the data concerning the role of exosomes during Dengue fever. As it appears, exosomes produced in response to DENV infection are also thought to facilitate the cell-to-cell transmission of viral particles by spreading various viral factors that enhance the susceptibility of cells and promote immune evasion. The various cargoes carried by exosomes could modulate immunity in T lymphocytes and platelets, and their action on endothelial cells may contribute to the pathophysiology of Dengue fever, notably vascular permeabilization during hemorrhagic forms of the disease [152].

The enclosure of viral factors in exosomes supports the Trojan Horse hypothesis, since the hijacking of exosomes by viruses seems an excellent means of promoting their incognito trafficking. Their capacity for prolonged circulation, as well as their nano-size and consistent shape give them access to extracellular spaces, thus explaining the efficient diffusion of their cargo. Moreover, exosomes can act as an effective platform to circumvent immune surveillance. As such, exosomes from APCs were shown to express membrane regulators of complement (CD46, CD55, and CD59) on their surface, preventing the activation of opsonin and coagulation factors [153]. Exosomes are thus protected from destruction by the complement cascade, warranting their stability and large-scale distribution in biofluids. In general, EVs have demonstrated their ability to regulate complement activity, albeit with variable effects, but contributing to the pro- and anti-inflammatory immune balance [154,155].

Finally, some infectious agents have the ability to corrupt exosome biogenesis to influence the localization of host-specific proteins, either by addressing them to exosomes, promoting their display at the cell surface, or concealing them in endosomal compartments. These mechanisms are mostly favorable to viruses and their ability to escape the immune system. For instance, HIV has been shown to promote the availability of its major entry receptors, CCR5 and CXCR4, on exosomes secreted by infected cells. The transfer of these receptors into recipient cells is associated with increased viral susceptibility [156]. Studies on EBV-infected cells have shown that they secrete exosomes with high levels of galectin-9, which triggers the apoptosis of CD4+ T cells and limits Th-1 responses [157,158]. Finally, the escape of herpes simplex virus 1 (HSV-1) from the immune system is due to its ability to hijack the addressing of viral peptide/HLA-DR complexes by promoting their sequestration in MVBs [159].

Fortunately, the ability of viruses to exploit exosomes is sometimes auspicious from a therapeutic standpoint. Indeed, oncolytic viruses have been introduced as a novel treatment strategy against malignancies. Interestingly, it has been observed that oncolytic adenoviruses could induce a systemic anti-tumor immunity, despite specifically and locally targeting malignant cells. The systemic effect was achieved when infected cells secreted exosomes loaded with viral genomes. Similar to the HCV pseudo-virions mentioned earlier, these exosomes could act as pseudo-viral particles and deliver the oncolytic genomes to other cancer cells. Their ability to circulate in body fluids and infiltrate tissues is in complete accordance with the systemic anti-cancer effect that has been documented for adenoviruses [160,161]. Although it was previously unknown that they could hijack exosomes, those findings may further explain how oncolytic viruses achieve the targeting of distant tumor cells, all the while evading the antiviral immune response.

The study of exosomes in infection biology provides many examples of their proviral effects, which mostly come as a consequence of exosome hijacking. However, many studies have contributed to cast a different light on exosomes, as they also appear to take part in the antiviral cellular responses and seem actively involved in the host response to pathogen intrusion.

### 3.2. Exosomes Contribute to Infection Resolution

Although their hijacking contributes to viral replication and dissemination, exosomes are canonically known for their aptitude to induce immune responses and their participation in immunomodulatory mechanisms. These functions have been extensively described in tumor processes [162]. The exosomes released during infection could play a similar role in modulating immune responses [163].

First, as stated above, exosomes from infected cells were shown to carry viral nucleic acids. This may lead to the stimulation of pattern recognition receptors in recipient cells (PRRs), such as cyclic GMP-AMP synthase stimulator of interferon genes (cGAS-STING) or RIG-I like receptors (RLRs), as demonstrated in the case of cancer-derived exosomes [164]. The downstream signaling pathways of these PRRs are part of the innate immune system, triggering the expression of inflammatory genes and type I interferon response, which are deleterious for pathogen replication and dissemination [165]. Such mechanisms were notably reported during HCV infection, illustrating the dual role played by exosomes that can have proviral and antiviral activities [166]. Secondly, exosomes can participate in antigen presentation, influencing the adaptive immune response [3]. This ability will be developed further below, as it contributes to the suitability of exosomes for vaccine development.

Supporting their ability to perform antiviral signaling, exosomes have been identified as carriers of various chemokines and cytokines important for infection resolution, like IL2, IL4, IL17, IL21, IL22, IL33, IFNα, IFNγ, TGFβ, TNFα, and other antiviral mediators [167,168]. This ability of cytokine encapsulation by exosomes is a feature shared by many viral infections and has even been achieved in vitro when stimulating keratinocytes with Poly(I:C) [169]. An exosome-mediated antiviral activity has been reported in the case of HBV [170] and HIV infections, through the delivery of INFα and other antiviral mediators [171]. Exosomes produced during DENV infection carry the interferon-induced transmembrane protein 3 (IFITM3), which promotes antiviral activities from cell to cell [172]. The cells that release such exosomes in response to infection are of several types, and this activity follows the stimulation of innate immunity pathways. In the case of HIV, IFNα-stimulated macrophages have been shown to produce exosomes enriched with the antiviral mediator apolipoprotein B mRNA editing enzyme catalytic polypeptide-like 3G (APOBEC3G), which hinders viral replication in recipient cells and confers them with resistance against HIV [171]. In addition, exosomes produced by TLR3-activated human brain microvascular endothelial cells were shown to carry mRNA and proteins of several interferon-stimulated genes (ISG), ISG15, ISG56, and Mx2, which could be transferred to recipient macrophages [173]. As final examples, exosomes derived from respiratory syncytial virus (RSV)-infected cells and carrying viral proteins were shown to trigger the production of various chemokines like monocyte chemoattractant protein-1 (MCP-1), interferon gamma-induced protein (IP-10), and (regulated upon activation normal T cell expressed and secreted) RANTES by monocytes [120]. Moreover, EVs released by DENV-infected macrophages that were characterized by the encapsulation of viral proteins and various miRNAs elicited an increased production of ICAM, TNF-α, IP-10, IL-10, RANTES, and MCP-1, triggering an antiviral defense program in endothelial cells [174].

As previously discussed, many viruses can exploit exosome biogenesis pathways to their advantage, and exosome production is often augmented during viral infections. Nevertheless, type I interferon was found to attenuate exosome secretion both in vitro and in vivo, thereby inhibiting their proviral effects. The mechanism is linked to the induction of ISG15, an ubiquitin-like protein that can bind to key actors of exosome biogenesis, notably TSG101, in a process termed ISGylation. This post-translational modification leads to the aggregation and degradation of TSG101, which is sufficient to significantly reduce exosome secretion by promoting the lysosomal degradation of MVBs [175,176].

Ultimately, an increasing number of studies have explored the capacity of exosomal miRNAs to modulate gene expression in recipient cells and improve their ability to reduce infection. For instance, exosomes derived from human trophoblast cells carry miRNAs inducing recipient cells to display higher levels of autophagy. Autophagy is a physiological process of protein and organelle turnover that is tightly linked with endosomal compartments and vesicular trafficking. As the production of viral progenies exclusively relies on the cell machinery, autophagy could impair the replication of various viruses through the sequestration of essential replication factors, the disruption of viral factories and replication platforms, or the addressing of newly synthesized viral particles to lysosomes [177]. Similarly, miR-483-3p, which is found in high levels in the serum of H5N1 influenza virus-infected mice, has been shown to mediate the expression of proinflammatory cytokines in vascular endothelial cells [178]. This could perhaps establish a systemic antiviral state that promotes the resolution of the infection. These examples illustrate how exosomes can globally alter cellular metabolism and provide cells with effective defense mechanisms to fight against viruses.

In conclusion, numerous studies attest that viral infections lead to a quantitative and qualitative modification of exosomes. Exosome signaling must be considered as a fully-fledged response, having a bidirectional regulatory effect on host–pathogen interaction. The hijacking of the exosome machinery enables pathogens to spread viral material and escape immune surveillance, whereas bona fide exosomes negatively regulate pathogen replication by transmitting antiviral mediators or immunomodulators and orchestrating immune defenses. The role of exosomes in facilitating the exchange of molecules and promoting their effective diffusion in tissues has inspired scientists to develop exosome-based therapies. These therapies have found applications in the fields of regenerative medicine, pharmacology, and vaccinology, as shall be discussed in the following section. Moreover, the quantitative increase in circulating exosomes observed during viral infections often correlates with the severity of the disease, as illustrated by flavivirus infections, notably with Dengue virus [139,179]. Exosomes are thus biomarkers of viral disease progression. This clinical use of exosomes to monitor the progression of a pathology is a new avenue being explored in human medicine. We shall provide a brief outline of this topic in the next section.

## 4. Exosomes as Biomarkers of Pathologies in Human Medicine

In the past few years, the use of exosomes as biomarkers and diagnostic tools has surged. Exosomes can be easily isolated from almost all body fluids, such as blood, urine, saliva, and even amniotic fluid [180]. The use of exosomes as indicators of physiological conditions is anecdotal. For example, they are a potential tool for fetal sex determination [181]. Most importantly, the quantity of circulating exosomes can serve as an indicator of pathological conditions. Exosome production has been reported to increase in the aftermath of a variety of cancers, in viral infections, as well as in acute kidney and cardiovascular diseases. Notably, this increase occurs simultaneously with the onset of the disease, reinforcing the potential of exosomes as early biomarkers [182].

In pathophysiological conditions associated with neurological diseases, viral infections, or cancer, the content of exosomes may undergo significant alterations. Many cancer studies now focus on circulating EVs in an attempt to establish correlations between the quantity or content of EVs and treatment responses, the presence of metastasis, or disease progression [183,184,185]. The analysis of EVs has emerged as a promising tool for cancer surveillance, surpassing traditional biopsy methods. EVs can be harvested in a minimally invasive manner from multiple biofluids, employing a technique known as liquid biopsy [182]. To meet clinical expectations for the use of exosomes as biomarkers, techniques for the ultrasensitive detection of exosomes from biopsy samples have been developed over the past ten years. Innovative systems based on optical, electrochemical, and electrical biosensors, widely discussed in several recent reviews, have enabled major advances in methods for the high-throughput screening, rapid isolation, and analysis of exosomes [186,187,188,189]. Another of their significant advantages lies in the easily detectable and analyzable nucleic acids they enclose. For example, double-stranded DNA molecules in exosomes can be used as predictive clinical biomarkers for cancer diagnosis and prognosis, as they may harbor mutations like KRAS and TP53 in pancreatic cancer [190]. The other candidates on the scene of exosomal biomarkers are miRNAs. Indeed, circulating exosomal miR-17-5p and miR-92a-3p have been associated with the pathological stage and grade of colon cancers [191], whereas miR-21 found in the cerebrospinal fluid can serve as a biomarker for the development of numerous cancers such as glioblastoma [192]. Likewise, circular RNAs (circRNAs), which proceed from mRNA splicing, may also serve this purpose. Their circular structure provides effective protection from exonuclease degradation, thus conferring greater stability compared to miRNAs [193]. Their relevance in cancer diagnosis has been investigated in pancreatic cancers [194].

Exosomes have been recently proposed as interesting tools for therapeutic drug monitoring. Nephrotoxicity is a notorious side effect associated with numerous drugs, including antimicrobials, anti-cancer therapies, and non-steroidal anti-inflammatory drugs. Their deleterious effects on the kidneys are currently monitored through the dosage of plasmatic biomarkers, which, unfortunately, lack sensitivity and specificity. Urinary exosomes could offer a good alternative. Differences in their surface proteins and miRNA content may correlate with specific renal diseases, thus facilitating diagnosis and adequate care [195].

Although exosomes present irrefutable advantages, their use in the clinical setting is yet to be popularized. Indeed, several subtypes of EVs may be found in different quantities in body fluids. If clinically relevant EVs are present in lower abundance, they might go unnoticed by detection methods, leading to potential false-negative results [196]. It should be emphasized that characterizing EVs remains a challenge. The prevailing methods often struggle to distinguish effectively between sub-populations of vesicles. Contaminants, such as viral particles or macromolecules, sharing similar characteristics and sizes, may not be adequately discriminated by the most commonly employed techniques [197]. Because of these limitations, experimental data should always be interpreted with great caution. Moreover, isolation methods and cargo analysis are often labor-intensive. Current techniques not only lack standardization but also require costly laboratory equipment that needs precise stewardship. This constitutes a lack of practicability that is currently incompatible with widespread use. As mentioned above, the development of ultrasensitive methods like microfluidic technologies and ongoing miniaturization efforts may address these challenges [198]. It is noteworthy that exosome-based diagnostic tools have already been trialed in clinical settings, demonstrating promising results in guiding clinicians to establish proper therapeutic interventions, as illustrated in this study on high-grade prostate cancer [199].

Exosomes appear as valuable biomarkers in the diagnosis of diseases and therapeutic drug monitoring. Nevertheless, their clinical potential could stretch even further. Their propensity to transport diverse molecules, coupled with their malleability, could propel exosomes into becoming nano-vectors of therapeutic molecules. In addition to their diagnostic features, exosomes could therefore acquire curative properties, as we shall explore in the next section.

## 5. Exosomes as Therapeutic Tools

Beyond their role as a diagnostic tool, EVs have received particular attention as highly promising therapeutic tools. Exosomes naturally produced by stem cells have demonstrated properties in tissue regeneration that could be used for therapeutic purposes. The unique ability of exosomes to carry a variety of cargo makes them valuable candidates for the targeted delivery of biologically active compounds. Thus, exosomes that have been specially designed to carry therapeutic molecules could be directed to specific cell types or tissues, offering a level of precision that is not always attainable with conventional drug delivery methods. In addition, exosomes have demonstrated their ability to cross biological barriers, enhancing their ability to reach target sites. Their natural role in intercellular communication further enhances their therapeutic utility. Exosomes show promise in a wide range of medical applications, from the treatment of cancer and infectious diseases to neurological disorders. Current research aims to unlock the full therapeutic potential of exosomes, paving the way for innovative and personalized approaches. This section will delve into the roles of exosomes in regenerative medicine and drug delivery.

### 5.1. Exosomes from Mesenchymal Stromal Cells in Regenerative Medicine

In recent decades, there has been a concerted effort in the development of regenerative medicine to heal tissues, restore their function, and hinder their senescence. As it is, many debilitating affections like cancer, chronic inflammatory, or neurodegenerative diseases remain incurable, and the absence of effective therapies has too often left the medical community powerless. Fortunately, the flourishing field of regenerative medicine offers several therapeutic prospects. Among these, cell-based therapy is being thoroughly investigated, and many cell types are currently examined for their regenerative properties, such as chimeric antigen receptor-T cells, pluripotent stem cells, and adult stem cells (ASCs) [200]. Mesenchymal stromal cells (MSCs) are subtypes of ASCs that can be identified by their surface markers. They reside in a broad variety of tissues, such as the bone marrow, skin, adipose tissue, umbilical cord, Wharton’s jelly, endometrium, and placenta. They are multipotent stem cells capable of differentiation into osteoblasts, chondrocytes, and adipocytes [201]. They were shown to display major immunoregulatory activities to control adverse inflammation and infection [202,203]. Their prominent contribution to immunomodulation and tissue homeostasis positions them as attractive regenerative cells.

However, cell-based strategies are facing several hurdles. First, MSCs constitutively express MHC class I molecules at their surface. Their inherent immunogenicity is problematic in the case of allogeneic injections and could trigger inflammation. Harvesting autologous MSCs could easily circumvent this issue, but the long cultivation process precludes their rapid implementation in patients suffering from acute diseases. Moreover, the use of MSCs has shown iatrogenic effects [204]. The injection of MSCs might alter their viability and result in the inoculation of necrotic byproducts, thereby exacerbating tissue inflammation [205]. Upon intravenous injection, transplanted MSCs are prone to senescence and, in some cases, have been reported as tumorigenic [206]. Finally, therapeutic MSCs are often collected from donor banks, posing ethical issues; their long-term culture could also favor the emergence of genetic and epigenetic mutations [207]. Consequently, the many challenges raised by cell-based therapies have incentivized scientists to design alternative solutions, namely, cell-free therapies.

Interestingly, the regenerative properties of MSCs are not primarily linked to their multipotency but rather proceed from their paracrine activity. The MSC secretome comprises soluble molecules (e.g., cytokines, chemokines, and growth factors) and EVs (e.g., exosomes) [208]. Therefore, MSC-derived EVs are of great clinical relevance, especially as their content is readily manipulable. Indeed, MSC priming refers to the in vitro culture of MSCs in a pre-conditioned medium. It dramatically influences the composition of their secretome, allowing for the production of disease-specific exosomes [209]. Accordingly, they may display anti-inflammatory [210], pro-angiogenic [211], anti-apoptotic, or anti-oxidative properties [212]. Cell-free therapies offer many advantages over cell-based therapies: a higher scale of production, a lesser risk of contamination with CMV and HSV, the possibility for long-term storage and easier transportation, and fewer ethical considerations [213]. Notably, MSC-derived EVs are believed to express fewer MHC molecules, which could explain their attenuated immunogenicity.

The biological effects of MSC-sourced EVs have been studied in vitro and in vivo. They have been primarily ascribed to exosomal miRNA. Their prominence in regulating physiological responses and their therapeutic relevance have recently led Dos Santos et al. to term them an MSC-EV-miRNAome [214]. When transported in MSC-derived exosomes, miRNAs are potent elicitors of biological responses, exerting immunomodulatory responses and anti-apoptotic or anti-proliferative actions. Their action results from their interference with key signaling pathways, thus triggering metabolic changes in the recipient cell. In ischemia-reperfusion injuries, exosomal miRNAs have been shown to attenuate tissue damage in the lungs and heart by inhibiting pro-apoptotic pathways [215,216] and gasdermin D-dependent pyroptosis [217]. In inflammatory diseases, miRNAs have shown immunomodulatory properties by polarizing immune cells, which preferably acquire an immunosuppressive phenotype, such as T regulatory cells, tolerogenic dendritic cells, and M2 macrophages [218]. Interestingly, chronic inflammatory diseases are often associated with the accumulation of fibrotic tissues. Fibrosis refers to a state of excessive extracellular matrix deposition, notably of collagen proteins, which progressively replace functional cells. Fibrosis is often considered to be an irreversible impairment. In the liver, hepatic stellate cells (HSCs) orchestrate collagen deposition and are the main actors in the progression of cirrhosis [219]. Remarkably, MSC-derived exosomal miRNAs could inhibit the Wnt-β-catenin pathway to hinder the proliferation of HSCs and therefore reduce liver fibrosis [220]. Finally, MSC-derived exosomes could be used as favorable substitutes for synthetic skin grafts in wound healing. By mitigating inflammatory pathways and up-regulating pro-angiogenic pathways, they could contribute to proper tissue reparation [205]. miRNAs may also diminish the excessive differentiation of myofibroblasts by inhibiting the TGF-β signaling pathway, thus tempering collagen deposition and tissue fibrosis [221]. The potentiality of MSC-EV-derived miRNAs is especially large, encompassing a broad range of diseases; an exhaustive delineation has been published elsewhere [222].

Despite their undeniable advantages, MSC-derived exosomes are far from being implemented as a routine treatment in clinical settings. As of now, current clinical trials have not yielded any commercially available exosome-based drug. This is compelling evidence that many issues remain unresolved. As underlined earlier, the production and isolation of exosomes lack a standardized procedure. Their intrinsic variability is commendable in the advent of personalized medicine, but in practice, it complicates the generation of reproducible batches, an inconvenience that is further accentuated by the coexistence of heterogeneous subpopulations of MSCs [223]. Hopefully, a recent secretome analysis has shown that peripheral blood mononuclear cells (PBMCs) could generate comparable batches, and secretome lyophilisates could remain stable over 6 months [224]. Therefore, the elaboration of an exosome-based drug that complies with the Good Manufacturing Practices posited by the World Health Organization appears achievable. However, many questions about their formulation, their administration route, and optimal therapeutic dose are left unanswered [225]. Additionally, despite their remarkable bioavailability, native exosomes spontaneously accumulate in the liver and spleen, where they are quickly removed from the blood flow by the mononuclear phagocyte system [226]. The artificial modification of exosome tropism could be an opportune solution to enhance their therapeutic efficacy. Finally, even though MSC priming can predictably influence the cargo composition, it does not allow for fine-tuned adjustments. As a consequence, contamination with undesirable molecules cannot be ruled out and compromises their innocuity.

Regarding their plasticity, biocompatibility, and low immunogenicity, exosomes are auspicious therapeutic tools. However, native exosomes are perfectible from a pharmacological standpoint. Therefore, scientists are attempting to engineer exosomes, designed to control both their composition and their precise targeting to cells and tissues. The ultimate goal would be to contain their pharmacological effect and make exosome-based therapies more amenable to clinical usage.

### 5.2. Engineered Exosomes for Drug Delivery

In pharmacology, molecular targets with hydrophobic pockets or located extracellularly are considered “druggable”, as they harbor accessible drug interaction sites. However, this statement entails that a significant majority, namely, 85% of targets, are classified as “non-druggable” [227]. This further implies that a large portion of molecular targets lack suitable characteristics for conventional drug development, potentially limiting the scope of drug discovery efforts. Hence, finding alternative drug delivery systems arises as a necessity.

Exosomes are recognized as natural carriers for drugs, holding great promise as drug delivery platforms for the treatment of various diseases, notably cancer. Delivering therapeutics encapsulated in membrane-bound vesicles enhances their bioavailability, circumvents the non-selective activity of drugs, and thereby reduces damage to normal cells. This holds particular relevance in the context of cancer chemotherapies [228]. Moreover, exosome-targeted vectorization enables the reduction of treatment doses, protecting cells from the cytotoxic side effects associated with high doses in diverse diseases. Their ability to cross the blood–brain barrier allows them to transport drugs to intracerebral regions, presenting a potential avenue for treating central nervous system (CNS) disorders [86].

Exploring the drug delivery potential of exosomes begins with the high-yield extraction of high-purity exosomes. Traditional methods often fall behind, but microfluidic-based techniques (e.g., immune-affinity-based micro-devices) provide a comprehensive solution by integrating multiple steps into a single device [229]. This approach enables the efficient extraction, analysis, and quantification of exosomes from limited clinical samples with high throughput and sensitivity. Integrating these advanced methods is crucial for achieving the fundamental isolation of high-purity, high-yield exosomes, a key step in developing effective engineered exosomes.

The choice of loading method is primarily guided by the nature of the molecule. Passive loading involves the coincubation of small lipophilic molecules (e.g., natural active principles such as curcumin and specific anticancer drugs) with exosomes or the exosome-producing cells [230]. When incubated with cells, the drug diffuses through the cell membrane and then interacts with the exosome biogenesis pathway, leading to its encapsulation within exosomes [231]. However, this method only applies to lipophilic molecules, capable of interacting with cell membranes and exosomes. Thus, loading large and hydrophilic proteins is considered challenging. To overcome this limitation, active loading methods are employed, which offer a more controlled and targeted approach. Techniques such as sonication, freeze-and-thaw cycles, and the application of saponin, a detergent that removes cholesterol from membranes, are commonly used. These methods permeabilize the membranes of cells and exosomes, enhancing the diffusion of specific molecules and ensuring a high loading efficiency of exosomal carriers [232]. Additionally, electroporation and transfection can be employed to load purified exosomes with proteins and nucleic acids, particularly miRNA and siRNA, often used for targeting cancer cells [233]. The incorporation of mRNA in this manner has raised the prospect of utilizing exosomes as vectors for messenger RNA vaccines. Furthermore, active methods encompass the use of genetically modified cells, which are considered an effective approach for loading drugs into exosomes.

By applying one of the methods mentioned above, several studies have demonstrated the ability of exosomes to transport drugs and treat specific diseases. Exosomes derived from macrophages transfected with a plasmid DNA encoding therapeutic proteins, such as catalase or glial cell-derived neurotrophic factor, have been used for the treatment of neurodegenerative disorders [234,235]. In another study, a co-delivery system was employed through exosome electroporation, relying on engineered exosomes carrying a chemotherapeutic drug (5-FU) and a miR-21 inhibitor oligonucleotide (miR-21i). It is noteworthy that miR-21 is widely recognized for its high expression in colorectal cancer cells. This approach demonstrated a significant improvement in the cellular uptake of exosomes and a significant reduction in miR-21 expression within 5-FU-resistant colorectal cancer cells. The systemic administration of exosomes loaded with both 5-FU and miR-21i led to a decrease in tumor growth in mice. In addition, this delivery approach reversed drug resistance against 5-FU and decreased the drug-induced cytotoxicity [236]. Another study mentions the use of exosomes derived from modified DCs that were loaded with the chemotherapeutic drug doxorubicin, through electroporation. Upon intravenous administration in tumor-affected mice, these engineered exosomes selectively delivered doxorubicin to tumor tissues, effectively inhibiting tumor growth without causing adverse effects to normal cells [233]. Finally, the therapeutic potential of exosomes modified to deliver natural substances with antioxidant and anti-inflammatory properties was examined. Thus, exosomes obtained from curcumin-treated macrophages displayed the capacity to suppress inflammation by activating the Nrf2/ARE pathway. Additionally, they enhanced the expression of molecules related to wound healing, stimulated angiogenesis, and accelerated the healing process in diabetic rats [231]. Moreover, curcumin-loaded exosomes derived from embryonic stem cells (MESC-exo^cur^) were generated through the simple incubation of exosomes and curcumin. MESC-exo^cur^ demonstrated the capacity to reduce inflammation and mitigate the expression of vascular ICAM and endothelial junction proteins. These exosomes have exhibited therapeutic potential in promoting neurovascular restoration following ischemia-reperfusion injuries in mice [230]. The promising outcomes of these studies not only underscore the versatility of EVs as drug carriers but also provide optimistic prospects for their therapeutic applications in diverse medical contexts.

Despite these positive aspects of using exosomes as drug vectors, their clinical application remains challenging. Difficulties in isolating and purifying exosomes pose significant hurdles. Moreover, the low efficiency in loading therapeutic cargoes, attributed to the intrinsic loading of exosomes with numerous proteins and nucleic acids, coupled with their rapid elimination from circulation, further complicates their clinical utility [237]. To address these challenges, consideration may be given to synthetic delivery systems, such as liposomes [238]. A notable advantage is the extended half-life of liposomes compared to exosomes, lasting several hours. However, synthetic drug delivery systems exhibit significantly lower targeting efficacy when compared to their natural counterparts [239]. Furthermore, liposomes primarily deliver drugs through passive accumulation in specific tissues, unless equipped with additional surface ligands [240]. In contrast, exosomes inherently possess targeting capabilities, potentially enabling drug delivery to specific cells [241]. This complexity sparks a debate on the effectiveness of using liposomes as a drug delivery system. In conclusion, while exosomes show promise as drug vectors, their clinical application still encounters significant challenges. These highlighted issues emphasize that the journey toward effective drug delivery systems still requires ongoing research and innovation.

## 6. Exosomes and Antigen Presentation: Perspectives in Vaccinology

While the therapeutic use of exosomes has become challenging, their unique ability to transport and present antigens to the immune system is of major interest in vaccinology, particularly in the field of virus control. Exosomes can display immunogenic viral antigens, acting as mediators that facilitate the recognition of these antigens by immune cells. The use of exosomes in antigen presentation has shown promise in stimulating robust immune responses. In addition, their natural origin and ability to interact with immune cells make exosomes attractive candidates for the design of innovative vaccine strategies. Current research focuses on exploiting exosomes as platforms for antigen delivery, exploring ways of modifying them to optimize the presentation of specific antigens and improve vaccine responses. By exploring the complex mechanisms of antigen presentation by exosomes, we are gaining valuable insights that contribute to the advancement of more effective and targeted vaccine approaches.

### 6.1. Exosomes and Antigen Presentation

A particular feature of exosomes that is receiving increasing attention is their immunomodulatory functions and their involvement in the presentation of antigens to the adaptive immune system (Figure 4). Exosomes derived from mature dendritic cells (DCs) were found to be enriched in MHC class I and II molecules, as well as ICAM-1 and CD86 [242]. Mature DC-derived exosomes are known to carry MHC–peptide complexes at their surface, and, in comparison with immature DC-derived exosomes, conferred mature DCs with a greater potency in priming naive T cells [243]. The presentation of pathogen-derived peptides by exosomes has been demonstrated in the case of bacterial and viral antigens. For example, it has been shown that the interaction of LPS-activated DCs with non-cognate activated T-cells induced changes in DCs morphological characteristics. This led to the release of exosomes carrying MHC class II-peptide, ICAM-1, and a high amount of miR155a. miR155a is a well-established central regulator of T-cell responses, possessing the capacity to activate antigen-specific CD8+ T cells [244]. Another study demonstrated that upon encountering *Escherichia coli*, DCs generated a substantial quantity of EVs, carrying MHC class II molecules and antigens derived from the previously phagocytosed bacteria. The underlying mechanism could be the inward budding of ILVs from the phagosomal membrane, allowing the internalization of bacterial byproducts. This investigation emphasized the potential effectiveness of such exosomes in presenting antigens and triggering an adaptive immune response when specifically packed with pathogen-derived peptides loaded onto MHC molecules [245].

Interestingly, exosomes that carry MHC–peptide complexes or antigens may act as indirect antigen presentation platforms, following internalization by other cells [3]. Indeed, migrating and resident DCs were demonstrated to exchange antigens through exosomes, thereby allowing DCs that were not directly exposed to the antigen to participate in T-cell priming. Such a phenomenon is termed cross-priming [246]. Following EV capture, antigens could either be driven into the endosomal pathway, subsequently degraded, and loaded onto MHC class II molecules; or they could be released into the cytoplasm, subsequently degraded, and loaded onto MHC class I molecules. Greater capacity for antigen presentation and efficient T-cell activation can thus be achieved by transferring antigens to DCs via exosome capture [247].

This ability to participate in the establishment of a specific immune response via EVs may also occur during viral infections. As discussed above, exosomes that are produced by infected cells are frequently loaded with viral factors, which can be displayed outwards, thus enhancing their immunogenicity. Concerning flaviviruses, exosomes originating from ZIKV- and TBEV (tick-borne encephalitis virus)-infected cells were found to display the viral envelope protein at their surface, suggesting their ability to induce an immune response [reviewed in: [248]]. Furthermore, our research has revealed that the non-structural protein-1 (NS1) of both DENV and ZIKV was found at the surface of EVs produced during infection [126]. However, the ability of these vesicles, which abundantly circulate in the body, to stimulate an immune response against the NS1 of these two flaviviruses has not been investigated yet.

Besides DCs and infected cells, various cell types may produce exosomes capable of triggering an immune response. For example, infection with the HTLV-1 retrovirus is accompanied by the detection of exosomes carrying viral proteins in the cerebrospinal fluid, where viral particles are undetected. PBMCs and CD4+CD25+ T cells appear as the principal producers of such exosomes, which contain the HTLV-1 Tax protein. Remarkably, these exosomes were shown to trigger a HTLV-1-specific immune response, and to induce the infiltration of HTLV-1 Tax-specific cytotoxic T lymphocytes (CTLs) into the central nervous system of patients [249].

Antigen-loaded exosomes produced by infected cells could also exert immunostimulatory actions, thereby contributing to the establishment of a proper immune response. A study has shown that the composition of exosomes isolated from COVID-19 patients correlated with the severity of the disease. Their proteomic analysis revealed a correlation between the protein repertoire and the immune response signature. In mild patients, exosomes were shown to carry SARS-CoV-2 Spike glycoprotein-derived peptides, along with MHC class II molecules, CD86, T-cell regulators, and antigen-processing factors. These exosomes exhibited the ability to effectively regulate antigen-specific CD4+ T-cell responses and induce IL-2 secretion in vitro. Conversely, in severe patients, exosomes were enriched in proinflammatory factors and were associated with unregulated inflammation, thrombus formation, and intravascular coagulation [250]. These results suggest that exosomes could participate in antigen presentation and act as key regulators of adaptive immune responses, thus significantly influencing the disease outcome. The roles of exosomes in eliciting adaptive immune responses are increasingly delineated, paving the way for the development of exosome-based vaccines.

### 6.2. Exosomes in Vaccinology

The inclusion of exosomes in the development of next-generation vaccines seems to be a promising application, particularly in the fight against emerging viral infections [251,252]. Due to their natural ability to convey immunomodulatory mediators, exosomes have been initially evaluated as potential nano-adjuvants in vaccine formulations. Indeed, to be effective, nano-adjuvants must be widely distributed. This is a clear advantage of exosomes over the various nanomaterials proposed for this role, such as graphene oxide, carbon nanotubes, various polymers, micelles, and liposomes [253]. The proof of concept for the use of exosomes as adjuvants was demonstrated during a trial of new vaccine strategies against HBV. Exosomes isolated from monocytic cells stimulated in vitro with lipopolysaccharide (LPS) endotoxin, that exhibited a non-specific immunostimulatory activity by themselves, were added to a vaccine formulation containing the classical HBV recombinant antigen (HBsAg). The subcutaneous administration of this combination to mice strengthened the immune response by increasing IFN-γ secretion [254].

Among the exosome-based vaccinal strategies against viruses, it has also been proposed to exploit the exosomes naturally produced during infection. This was the case for EBV after it was discovered that the infection of B cells induced the production of exosomes with the viral glycoprotein 130 (gp130) exposed at their surface. These exosomes have demonstrated interesting antiviral and vaccine properties against EBV in vitro. Exosomes exposing g130 were shown to interact with CD21, a marker that is consistently expressed by B cells and follicular DCs. Then, the interaction of exosomal gp130 with uninfected B-cells protected them against EBV infection, due to the saturation of CD21 receptors which precluded the adsorption of viral particles. Moreover, it has been hypothesized that the exosomal delivery of gp130 to follicular DCs or uninfected B cells may trigger an effective adaptive response against this antigen, which could be compatible with the genesis of an EBV-specific immunological memory [255].

However, the development of exosome engineering techniques has paved the way for several new vaccine strategies. These are based firstly on the antigen-presenting capacity of exosomes, the content of which can be modified to turn them into tools for stimulating immunity. They also take advantage of the bioavailable nanocarrier characteristics of exosomes to vectorize and deliver nucleic acids, providing alternative solutions to the development of messenger RNA vaccines. For the first approach, exosomes offer several advantages, including a more stable conformation for vectorized antigens, compared with other formulations. Their broad distribution through their ability to circulate in biological fluids, enables exosomes to reach distant targets. Moreover, because of the adhesion molecules present on their surface, they offer the most effective biomimetic platforms to maximize association with antigen-presenting cells.

The engineering of exosomes for vaccine purposes is nonetheless challenging, at least as challenging as the drug delivery design discussed in Section 5.2. In order to make them suitable vaccination platforms, the assembly of targets of interest (DNA, RNA, or proteins) at the surface or within EVs is required. Several methods have been explored to date, each more or less efficient depending on the nature of the target to be incorporated.

### 6.3. Exploring Strategies for Loading Exosomes with Antigens

As highlighted earlier, exosomes serve as effective antigen-presenting devices, showcasing their capability to stimulate the immune system. Consequently, they are emerging as promising candidates for enhancing vaccine strategies. This section delves into various manufacturing techniques employed to precisely load exosomes, ensuring the production of exosomes carrying the desired antigenic determinants. These exosomes are likely to elicit a robust immune response.

Strategies for directing immune modulators or antigens to exosome membranes have primarily revolved around the concept of “exosome display,” a technique developed nearly two decades ago [256]. The aim is to introduce a transgene into cells, enabling the expression of a chimeric protein addressed to the membrane of ILVs during their biogenesis. The transgene construct results from the fusion of sequences from different origins: one encodes the antigen to be exposed, a second encodes a signal peptide promoting the membrane insertion of the protein, and a third encodes an exosome targeting domain (EDT). The C1C2 domains of lactadherin, known for their ability to direct the secretion of lactadherin-containing exosomes, are among the most widely used EDTs (illustrated in Figure 5A). Positioned at the C-terminus of the fused sequence, the C1C2 domain directs the entire protein into secreted exosomes and places the N-terminal region outward at the exosome surface. This method has been successfully applied, using the monomeric enhanced green fluorescent protein (EGFP) as the antigen, fused to the C1C2 domain. This approach made it possible to extract exosomes produced by cells transfected in vitro with this plasmid construct, demonstrating the display of EGFP at the exosome surface. Most importantly, the intramuscular and intranasal administration of purified exosomes, in combination with a conventional vaccine based on an adenoviral expression vector for the EGFP antigen, significantly amplified the humoral immune response against EGFP in mice [257].

Another approach used for addressing antigens to exosomes involves tetraspanins. As mentioned in Section 2.1, these transmembrane proteins participate in the cargo sorting process that occurs during ILVs formation. Thus, they are considered highly suitable for displaying antigens at the surface of exosomes, either bound to their extravesicular or intravesicular domains [258]. A study employing several types of genetic constructs, consisting of chimeric fusions of the VSV-G gene fused to tetraspanin genes (CD9, CD63, or CD81), confirmed the effectiveness of this exosome manufacturing system. However, the collected exosomes revealed variable physicochemical properties, as well as differences in surface protein profiles and cellular uptake characteristics [259]. Additionally, some other transmembrane proteins like integrins, Prostaglandin F2 Receptor Negative Regulator (PTGFRN), brain abundant membrane attached signal protein-1 (BASP1), and Lysosome-associated membrane protein-2b (LAMP-2B), were shown to be able to bind antigens to the exosomal membrane, making them de facto proteins of interest for scaffold platforms of antigen presentation (illustrated in Figure 5B) [260,261]. Numerous studies have reported the widespread use of LAMP-2B-based engineering for loading proteins of interest onto the surface of exosomes. Beyond its role in antigen exposure, this method has been employed to confirm the targeting capabilities of exosomes, depending on the selected protein for display. Thus, LAMP-2B constructs fused with a rabies viral glycoprotein peptide, known to direct rabies virus neurotropism, effectively enabled the production of exosomes that reached the brain [262]. If a ligand is yet to be identified for a tissue- or cell-specific receptor, protein libraries can be screened or biopanned using high-throughput phage display methods. Upon finding a peptide with adequate affinity for the target receptor, it can be fused to LAMP-2B and expressed on exosomes. Subsequent in vivo experiments can be performed to assess whether exosomes display the expected tropism. This strategy has been adopted to specifically target exosomes to hepatic stellate cells, which are relevant targets in liver fibrosis [263].

Numerous adaptations of exosome display, based on the expression of transgenes, have been developed. As previously discussed, specific viral proteins exploit the exosome biogenesis pathway to facilitate virus budding. It has been shown that transmembrane subunits of viral envelope proteins, including those of HIV, can interact with components of the ESCRT machinery, such as TSG101 [264]. These properties, which are found in the proteins of various viruses (e.g., the VSV-G protein or the Nef protein of HIV), have been exploited to direct proteins of interest to exosomes [265]. In order to assess the exosomal addressing capabilities of viral proteins, a vector encoding the CD8 ectodomain, used as a neutral reporter, fused with the cytosolic domain of the transmembrane envelope protein (CDTM-Env) derived from the bovine leukemia virus (BLV) was employed. Cells transfected with this construct successfully produced exosomes expressing the chimeric protein. This finding indicates the potential utilization of specific domains within certain viral proteins for directing antigens into exosomes [266]. Exosome engineering likely needs to be adapted to the specific type of antigen intended for expression. Comparisons of reproducibility and yields across various exosome display systems, depending on the constructs and techniques used to introduce transgenes into cells, should be conducted to identify the optimal technique.

In addition to surface display engineering of exosomes, another strategy termed “cloaking” has been developed to make exosomes presenting scaffolds for proteins of interest. This approach involves the ex vivo modification of exosome surfaces to enhance their targeting precision toward recipient cells. The proof of concept was put forth using cardiosphere-derived cells exosomes, cloaked with a phospholipid membrane anchor and streptavidin. The coating with streptavidin molecules enables the binding of any biotinylated molecule at the surface of exosomes, including tissue-specific antibodies and homing peptides. This technology could be beneficial for the development of exosome-based vaccines, as it would allow the precise targeting of antigens to APCs [267].

Furthermore, the lipid bilayer of exosomes holds substantial potential for anchoring lipid conjugates (Figure 5B). It is composed of different lipids like cholesterol, ceramide, and phosphatidylethanolamine derivatives that serve as lipid anchors in regions called lipid rafts, ensuring the attachment of functional entities to the exosome membrane. A recent study demonstrated that the incubation of fluorophore-bound dioleoyl phosphatidylethanolamine (DOPE) with EVs leads to its effective anchoring to the EV membrane. This approach could be used to activate the immune response when the fluorochrome is replaced by an antigen of interest, which could subsequently be used as in a vaccine [268]. Furthermore, lipids of the exosomal membrane can be covalent protein-anchoring systems, which is the case for glycosylphosphatidylinositol (GPI)-anchored proteins. Thus, several studies have reported the possibility of efficiently addressing proteins of interest on the surface of exosomes by introducing in cells transgenes that drive the expression of chimeric (GPI)-anchored proteins [269,270].

The collective findings suggest that loading exosomes with tailored antigens through different strategies appears to be a viable approach for the development of vaccines targeting a range of diseases and pathogens, including flaviviruses.

### 6.4. Ongoing Exosome-Based Vaccine Strategies against Viruses

Thanks to the wide range of available techniques for engineering exosomes into antigen-presenting platforms, exosome-based candidate vaccines against several viruses have been researched and have undergone several phases of validation of their suitability. A trial was conducted with exosomes modified to carry HCV full-length NS3 protein. The modified exosomes proved immunogenic when injected into mice, which was confirmed by the detection of NS3-specific memory CD8+ T cells [271]. Anticoli et al. have also demonstrated the efficacy of engineering exosomes with DNA vectors expressing stable fusion products of viral antigens fused to an exosome-anchoring protein. Assayed viral antigens included those of the Ebola virus, influenza virus, Crimean–Congo hemorrhagic fever, and West Nile virus. Upon injection in mice models, these vectors have been shown to elicit a potent antigen-specific CD8+ T-cell response, which was further supported by their cytotoxic activity against antigen-expressing cells [272].

More recently, in the face of the global COVID-19 crisis, there has been a flurry of research into innovative vaccination strategies exploiting the exosome approach [273], [274]. Two of these strategies are summarized in Figure 6. One strategy, following the popularization of mRNA vaccines, strove to turn exosomes into mRNA vectorization tools. The use of exosomes to vectorize mRNAs for vaccine purposes has been successfully tested and shown to promote an immune response [275]. Thus, for example, Popowski et al. developed a system based on lung-derived exosomes carrying SARS-CoV-2 Spike mRNA. These exosomes have shown extensive distribution in the bronchioles and parenchyma of mice. In addition, they elicited significant immunoglobulin G (IgG) and secretory IgA (SIgA) production, demonstrating the relevance of such an approach for obtaining a good adaptive immune response [276]. Other work showed that bovine-milk-derived exosomes loaded with mRNA encoding the SARS-CoV-2 receptor-binding domain (RBD) stimulated the production of neutralizing antibodies against RBD in mice [277].

It is noteworthy that research on exosome-based vaccines against coronaviruses has been ongoing since 2007 [278]. Thus, it was demonstrated that exosomes containing the Spike (S) proteins of SARS-CoV, the transmembrane and cytoplasmic domains of which were replaced with those of the vesicular stomatitis virus glycoprotein (VSV-G), induced high levels of neutralizing antibodies [278]. In the context of the COVID-19 pandemic, this strategy of modifying exosomes as antigen-presenting platforms has been relaunched by a growing number of start-ups and private or semi-private biotech companies worldwide, interested in developing extracellular vesicle-based therapies and virus- and adjuvant-free vaccines based on recombinant exosomes [279]. In this respect, an extremely promising vaccine candidate that has passed pre-clinical trials proposes the inhalation delivery of pulmonary exosomes, which carry the recombinant SARS-CoV-2 RBD. The process of this second strategy, related to the use of exosomes as an antigen-presenting device, is described in Figure 6. The engineered and purified exosomes had the advantage of being remarkably stable after lyophilization. They have been shown both to enhance RBD retention in the airways and lung parenchyma, and to induce the production of RBD-specific IgG antibodies in mice. They also enabled a mucosal IgA response and an effective T-cell response that was able to clear the virus in a challenge test [280]. To the best of our knowledge, this promising work has given new momentum to ongoing research by start-ups into exosome-based vaccine strategies against the Chikungunya, Zika, Dengue, and West Nile viruses.

Although vaccines confer the host with durable immunity, pathogens are notorious for evading the immune system through antigenic variation, a challenge that is usually overcome with the use of cocktails and multi-epitope vaccines. In this regard, exosomes could be engineered to present multiple antigens. During the SARS-CoV-2 epidemic, all commercially available vaccines targeted the S glycoprotein, which was consequently subjected to a strong selective pressure. The rapid mutation rate gave rise to numerous variants that could escape the vaccine-induced immunity, notably omicron variants. In contrast, other viral proteins, such as the nucleocapsid (N) protein, are less prone to mutations. In order to broaden immunity against coronaviruses, Cacciottolo et al. have devised a bivalent exosome-based vaccine against delta SARS-CoV-2. Using cell lines that stably expressed both S and N proteins, they produced S- and N-laden exosomes. Upon injection in mice and rabbits, these exosomes elicited a robust T-cell response and significant titers of neutralizing antibodies. Strikingly, these antibodies demonstrated cross-neutralization abilities against omicron variants. Interestingly, exosomes were injected without adjuvants, and low amounts of viral proteins sufficed to produce an efficient response. Finally, the vaccine was regarded as safe and only two shots were required to achieve immunity [274].

The aforementioned strategies exemplify the promising applications of exosomes in the field of vaccinology. However, exosome-based vaccines against viruses are still in their infancy and, so far, they have not produced any widely approved formulations. Further studies must be performed to assess their efficacy and innocuity, but current developments are nonetheless encouraging.

## 7. Challenges and Issues

Exosomes show promise in advancing medical treatments, such as cell-free therapies, drug nanocarriers, or vaccines, offering targeted and efficient options. However, there are still many difficulties to overcome.

Regarding exosome-based vaccines, a challenge arises in the formulation process, which requires the production of consistent batches of exosomes, each containing the adequate levels of antigen necessary for effective protection [281]. Moreover, a large-scale and efficient source of exosomes, such as high-throughput cellular systems, must be considered and carefully thought-out, to meet the growing demand. This consideration becomes particularly significant in light of the recent SARS-CoV-2 epidemic, emphasizing the urgency for swift and sustained vaccine availability to achieve effective herd immunity. Simultaneously, it is crucial to manage the immunogenicity of exogenous components, namely, those derived from the cell line used to produce exosomes. Another limitation is the heterogeneity of exosomes, which contain a large number of diverse proteins and materials from host cells, making it challenging to customize their cargoes compared to liposomes [282]. While immunogens and adjuvants can be easily packaged in liposomes, incorporating desired components into exosomes has proven challenging at times [283]. In addition to the advantages afforded by standardized liposomes for the delivery of therapeutics, the “exosome” strategy is being challenged by other strategies for the custom development of delivery systems based, for example, on bacteriophage virus-like particles (VLPs) [284] or other biomimetic platforms [285]. Future research on exosomes for vaccine purposes will have to establish a comparison with these other types of nano-platforms, particularly in preclinical studies. Furthermore, when exosomes are injected into a patient, they may compete with their endogenous counterparts, potentially affecting the desired therapeutic effects [282], or interfering with intercellular communication networks. They may also bind to competitive ligands instead of cell receptors [286]. Exosomes harvested from DCs could hold the best vaccine potential, as they naturally express antigen-presentation molecules. This strategy has already proven its relevance in oncology, with DC exosome-based vaccines that have passed both preclinical and clinical phases, demonstrating good tolerance and safety [287,288]. On the downside, the many clinical trials carried out in recent years with exosome-based cancer immunotherapies have not always produced the expected results, prompting further research into the immune mechanisms controlled by exosomes in vivo [289].

Finally, ethical considerations surrounding safety, informed consent, and unpredictable consequences must be addressed. Rigorous testing, transparency in research, and clear patient communication are essential. Balancing innovation with ethical standards is paramount to harness the full potential of exosomes in medicine. The evolution of guidelines to ensure responsible and beneficial applications is strikingly evident.

## 8. Concluding Remarks

The global health crisis caused by SARS-CoV-2 has reinforced the need to diversify vaccine strategies against viral infections. This, in turn, has renewed interest in exosomes as particularly attractive nanocarriers. We therefore set out to review the literature on the biology of exosomes, which mediate signaling between cells and play a key role in both physiological and pathological processes. We pointed out their considerable relevance as potential biomarkers and diagnostic, and therapeutic tools in contemporary medicine. Their role in viral infections is increasingly studied, and we have pointed out that it is often dual, complicating the prospects for their therapeutic usage. However, significant advances in the engineering of these vesicles for the custom loading of proteins or RNAs confirm that they are interesting nano-vectors of new therapeutic molecules. Likewise, exosomes could benefit from the innovative design of vaccines, unveiling novel strategies for viral infection control. The recent demonstration of their efficacy in preclinical models strengthens their use as original platforms for antigen presentation or mRNA vectorization systems. However, uncertainties remain on several aspects such as the choice of exosome-producing cells, engineering methods, standardized production, the safety of administration, and ethical considerations. Nevertheless, this paves the way for the exploration of new approaches to control flavivirus infections, such as DENV or ZIKV, for which vaccine design has become challenging. The transition to clinical trials will, of course, be necessary to validate these approaches and, hopefully, propel exosome-based therapies from bench to bedside.

## Figures and Tables

**Figure 1 vaccines-12-00280-f001:**
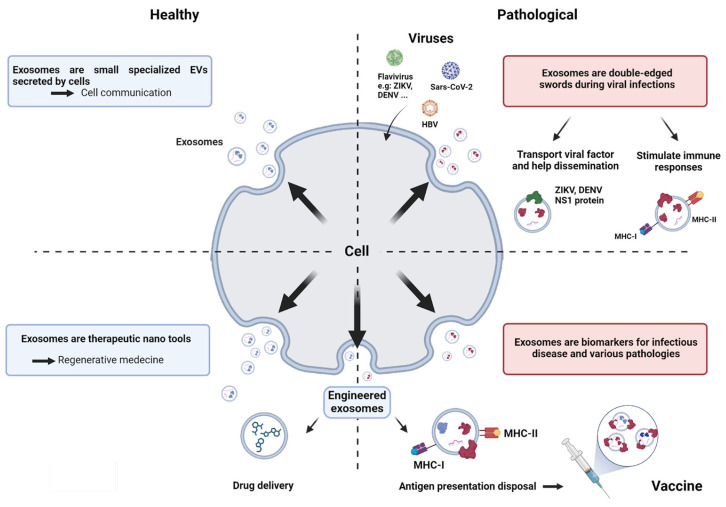
Overall diagram outlining the different aims of the present review, with a particular focus on exosomes in the context of viral infection, as antigen presentation disposals, and as antigen-presenting devices in vaccine strategies.

**Figure 3 vaccines-12-00280-f003:**
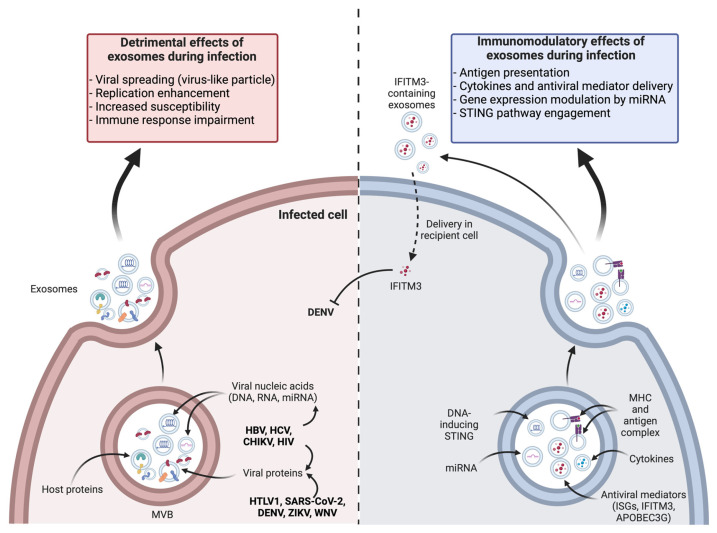
Exosomes are double-edged swords during infection. On the one hand, exosomes may have detrimental effects during infection by promoting viral replication, impairing immune response, and increasing susceptibility, ultimately leading to enhanced viral spreading. On the other hand, exosomes can participate in controlling infection through their immunomodulatory effects, including antigen presentation, the delivery of cytokines and antiviral mediators, miRNA delivery, and the engagement of the stimulator of interferon genes. Created with Biorender.com.

**Figure 4 vaccines-12-00280-f004:**
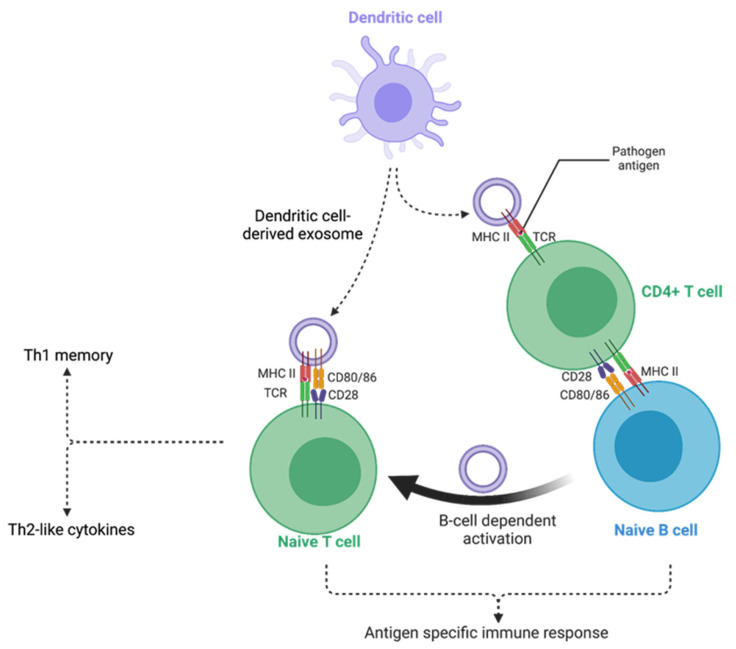
Antigen presentation by exosomes. Having been primed with non-self antigens, dendritic cells may emit exosomes wherein antigens have been enclosed. Upon interaction with naïve dendritic cells, exosomes may deliver the non-self antigens, in a process called cross-priming. Although recipient dendritic cells have never been directly exposed to them, they can now present the exosome-derived antigens to CD4+ T cells and CD8+ T cells. CD4+ T cells will further activate naïve B cells and induce clonal expansion and differentiation. Cellular immunity response will thus be set up, leading to Th1 memory and Th2-like cytokines. A particularity of exosome and antigen presentation is that B-cell-derived exosomes can present allergen peptides and activate allergen-specific T cells.

**Figure 5 vaccines-12-00280-f005:**
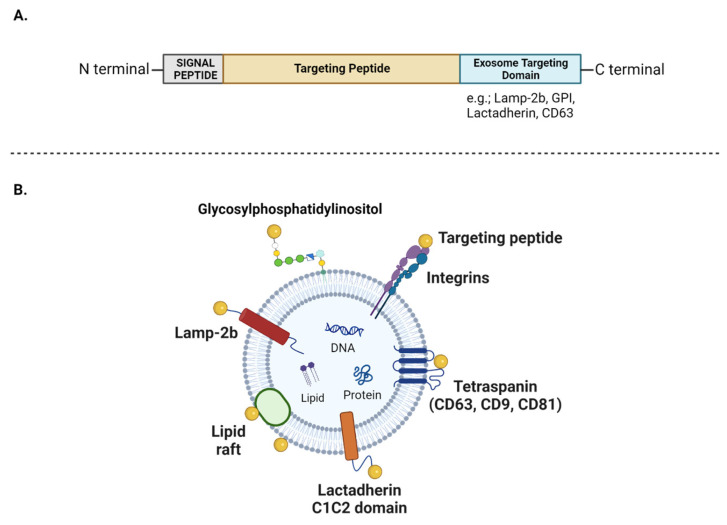
Various strategies of exosome display: Target peptide presentation and surface binding mechanisms. (**A**). A fusion gene model for targeting recombinant proteins to exosomes. To display proteins at the surface of exosomes, cells undergo transfection with plasmids encoding a chimeric protein. This chimeric protein exhibits a signal peptide present in the N-terminal region, the antigen of interest, and an exosome targeting domain generally positioned in the C-terminal region if the peptide is intended for display at the exosome surface. (**B**). Schematic diagram showing the presentation of a target peptide via different transmembrane domains of the exosome. Exosomes exhibit the presentation of target peptides on their surface through diverse exosomal transmembrane domains, showcasing the outcome of the exosome display strategy. Moreover, target peptides can bind to the exosome surface either through the cloaking strategy or by associating with the lipid rafts spanning the exosome membrane.

**Figure 6 vaccines-12-00280-f006:**
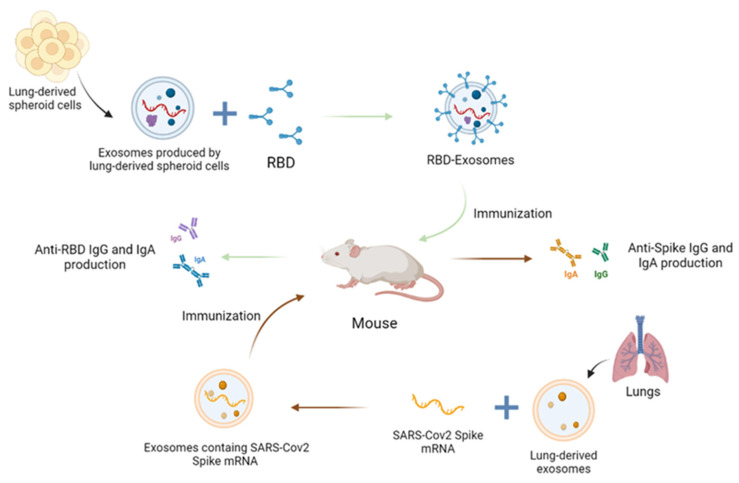
Exosomes modified according to two strategies’ have been proposed as candidate vaccines against SARS-CoV-2. They are either designed to carry viral mRNA or engineered to expose a viral antigen, such as the Spike RBD. Both of these candidate vaccines demonstrated protection against SARS-CoV-2 infection in mice.

## Data Availability

Data sharing is not applicable.

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
