# Peer review of "Exosome-Mediated Antigen Delivery: Unveiling Novel Strategies in Viral Infection Control and Vaccine Design"

_vaccines, 2024, doi:10.3390/vaccines12030280_

Round 1

Reviewer 1 Report (Previous Reviewer 3)

Comments and Suggestions for Authors

The manuscript has thoroughly been revised, open questions have been answered, valuable explanations and additional information have been added to the manuscript.

Comments on the Quality of English Language

Minor editing of English language required.

Author Response

We would like to thank you for your positive comments and we are pleased that you find our review clear and that it answers the various questions you raised.

Reviewer 2 Report (New Reviewer)

Comments and Suggestions for Authors

The review “Exosome-mediated Antigen Delivery: Unveiling Novel Strategies in Viral Infection Control and Vaccine Design” effectively highlights the diversity and complexity of exosomes in communication, homeostasis, and infectious processes including cancer. It distinguishes the evolving understanding of exosome composition and the dual role in viral infections. The review provides details on the molecular makeup and environmental influences on exosomes.

Major comments:

-        The review could benefit from more critical analysis of potential limitations, gaps, or controversies in the current understanding of exosome biology and their roles in infections.

-        The review lacks in-depth discussion of potential limitations, controversies, and recent findings like the role of bacteriophages in exosome functions which are supposed to be a critical factor in exosome mediated functions and must be considered equally important factor in immune regulation! Please refer PMCID: PMC9878357, PMCID: PMC8473360,

-        While the review effectively synthesizes existing knowledge, the rapidly evolving nature of this field may require updates to incorporate recent research findings for a more comprehensive overview.

-        The review could benefit from elaborating on the current status of clinical trials and practical applications.

-        Further exploration of ethical concerns surrounding exosome use in research and medicine is needed.

-        Discussing potential future research directions would provide a forward-looking perspective.

Comments on the Quality of English Language

It is fine

Author Response

The review “Exosome-mediated Antigen Delivery: Unveiling Novel Strategies in Viral Infection Control and Vaccine Design” effectively highlights the diversity and complexity of exosomes in communication, homeostasis, and infectious processes including cancer. It distinguishes the evolving understanding of exosome composition and the dual role in viral infections. The review provides details on the molecular makeup and environmental influences on exosomes.

We thank the reviewer for his comments. We've indicated the changes made in green in the revised version.

Major comments:

1-        The review could benefit from more critical analysis of potential limitations, gaps, or controversies in the current understanding of exosome biology and their roles in infections.

We recognize that to this day there are still gaps and limitations in the current understanding of exosome biology.

We therefore address the issue of variability in exosome content, which potentially fuels the difficulty and controversy of exosome analysis and their physiological effects which are themselves complicated by the difficulties of determining the fate of exosomes in vivo.

In section “2.2. Exosomes composition”

Line 255: These uncertainties regarding exosome content arise from advances in techniques for separating small vesicles and the growing availability of powerful analytical methods, such as mass spectrometry, for studying their proteomes [63]. 

In section “2.3. Exosomes biological activities according to their mode of interaction with recipient cell”

Line 438: In conclusion, the vast amount of data on the biological effects of exosomes reveals the multifaceted nature of exosome-mediated cellular communication. The substantial development of research into their activities in the field of cancer confirms their key role in the progression of these diseases. Meanwhile, this intensive research has also revealed the difficulty of exploring the fate of exosomes in vivo, leading to potential controversies and limitations. Despite the often ambiguous and challenging-to-delineate activities of exosomes, harnessing their knowledge could help find sustainable applications, including innovative diagnostic approaches and targeted therapeutic interventions. 

We further provide additional details on the improved isolation techniques recently developed, supported by new references from 2024.

Line 707: To meet clinical expectations for the use of exosomes as biomarkers, techniques for ultrasensitive detection of exosomes from biopsy samples have been developed over the last ten years. Innovative systems based on optical, electrochemical and electrical biosensors, widely discussed in several recent reviews, have enabled major advances in methods for high-throughput screening, rapid isolation and analysis of exosomes [186], [187], [188], [189]. 

We would like to underline that controversies concerning the pro- or anti-tumor activity of exosomes have been extensively discussed all along the section 2.3. on Exosomes biological activities according to their mode of interaction with recipient cell

Concerning the research carried out to establish the functions of exosomes during viral infection, we have highlighted and discussed, with factual elements from published data, the duality of the effects currently described. We propose a clear critical analysis in two successive sections: 3.1 deciphering their role as contributors to viral dissemination; 3.2 deciphering their role as contributors to infection resolution. Given these ambiguous roles in infectious pathology, and to meet your expectations, we're expanding the discussion in the light of recent research :  line 504 and 673.

 line 504 “This ability has led to a Trojan Horse hypothesis that exosomes actively contribute to infectious pathogenesis, helping some viruses to evade immune system recognition and sometimes facilitating viral persistence [135].”

 line 673 “In conclusion, numerous studies attest that viral infections lead to a quantitative and qualitative modification of exosomes. Exosome signaling must be considered as a fully-fledged response, having a bidirectional regulatory effect on host-pathogen interaction. Hijacking of the exosome machinery enables pathogens to spread viral material and escape immune surveillance, whereas bona fide exosomes negatively regulate pathogen replication by transmitting antiviral mediators or immunomodulators and orchestrating immune defenses.”

However, in the case of infectious pathologies, systematic quantitative changes have prompted scientists to focus on exosomes as biomarkers. We examine this point in more detail with reference to a study on the subject from early 2024.

 line 683:

Moreover, the quantitative increase of circulating exosomes observed during viral infections often correlates with the severity of the disease, as illustrated by flaviviruses infections, notably with Dengue virus [179], [139]

In section 4 “exosomes as biomarkers”, we emphasize these limitations.

Line 732 :

“Although exosomes present irrefutable advantages, their use in the clinical setting is yet to be popularized. Indeed, several subtypes of EVs may be found in different quantities in body fluids. If clinically relevant EVs are present in lower abundance, they might go unnoticed by detection methods, leading to potential false-negative results [194]. It should be emphasized that characterizing EVs remains a challenge. The prevailing methods often struggle to distinguish effectively between sub-populations of vesicles. Contaminants, such as viral particles or macromolecules, sharing similar characteristics and sizes, may not be adequately discriminated by the most commonly employed techniques [197]. Because of these limitations, experimental data should always be interpreted with great caution.  Moreover, isolation methods and cargo analysis are often labor-intensive, and current techniques not only lack standardization but also require costly laboratory equipment that need precise stewardship; a lack of practicability that is currently incompatible with widespread use. As mentioned above, the development of ultrasensitive methods like microfluidic technologies and ongoing miniaturization efforts may address these challenges [195]. It is noteworthy that exosome-based diagnostic tools have already been trialed in clinical settings, demonstrating promising results in guiding clinicians to establish proper therapeutic interventions, as illustrated in this study on high-grade prostate cancer [196].

We also discuss more in depth some controversial aspects of the benefits of focusing on exosomes in several sections of the review:

Section 5.2 

Line 946, comparing strategies using other vectorization systems and on the many issues yet to be clarified:

Despite these positive aspects of using exosomes as drug vectors, their clinical application remains challenging. Difficulties in isolating and purifying exosomes pose significant hurdles. Moreover, the low efficiency in loading therapeutic cargoes, attributed to the intrinsic loading of exosomes with numerous proteins and nucleic acids, coupled with their rapid elimination from circulation, further complicates their clinical utility [237]. To address these challenges, consideration may be given to synthetic delivery systems, such as liposomes [238]. A notable advantage is the extended half-life of liposomes compared to exosomes, lasting several hours. However, synthetic drug delivery systems exhibit significantly lower targeting efficacy when compared to their natural counterparts [239]. Furthermore, liposomes primarily deliver drugs through passive accumulation in specific tissues, unless equipped with additional surface ligands [240]. In contrast, exosomes inherently possess targeting capabilities, potentially enabling drug delivery to specific cells [241]. This complexity sparks a debate on the effectiveness of using liposomes as a drug delivery system. In conclusion, while exosomes show promise as drug vectors, their clinical application still encounters significant challenges. These highlighted issues emphasize that the journey toward effective drug delivery systems still requires ongoing research and innovation.

In the discussion of the section 6.2 

Line 1081 :

 However, the development of exosome engineering techniques has paved the way for several new vaccine strategies. These are based firstly on the antigen-presenting capacity of exosomes which content can be handled to turn them into tools for stimulating immunity. They also take advantage of the bioavailable nanocarrier characteristics of exosomes to vectorize and deliver nucleic acids, providing alternative solutions to the development of messenger RNA vaccines. For the first approach, exosomes offer several advantages, including a more stable conformation for vectorized antigens, compared with other formulations. Their broad distribution through their ability to circulate in biological fluids, enables exosomes to reach distant targets. Moreover, because of the adhesion molecules present on their surface, they offer the most effective biomimetic platforms to maximize association with antigen-presenting cells.

The engineering of exosomes for vaccine purposes is nonetheless challenging, at least as challenging as the drug delivery design discussed in section 5.2. In order to make them suitable vaccination platforms, the assembly of targets of interest (DNA, RNA or proteins) at the surface or within EVs is required. Several methods have been explored to date, each more or less efficient depending on the nature of the target to be incorporated. 

And in the concluding remarks Line 1337:

The recent demonstration of their efficacy in preclinical models strengthens their use as original platforms for antigen presentation or mRNA vectorization systems. However, uncertainties remain on several aspects such as the choice of exosome-producing cells, engineering methods, standardized production, safety of administration and ethical considerations. Nevertheless, this paves the way for the exploration of new approaches to control flavivirus infections, such as DENV or ZIKV, for which vaccine design has turned challenging. The transition to clinical trials will of course be necessary to validate these approaches and, hopefully, propel exosome-based therapies from bench to bedside.  

2-        The review lacks in-depth discussion of potential limitations, controversies, and recent findings like the role of bacteriophages in exosome functions which are supposed to be a critical factor in exosome mediated functions and must be considered equally important factor in immune regulation! Please refer PMCID: PMC9878357, PMCID: PMC8473360,

We thank the reviewer for pointing out that recent discoveries would reveal a role for bacteriophages in exosome function, but we didn’t find any published data establishing a direct link. The references you provide do not mention any effect of bacteriophage infection on exosome content or function. However, the references you have recommended have enabled us to develop a discussion on research perspectives in the field of exosomes as antigen nanocarriers, in comparison with other bio-compatible platform production systems such as the VLPs bacteriophages described in the 2 publications.

Line 1313: This strategy has already proven its relevance afforded by standardized liposomes for the delivery of therapeutics, the "exosome" strategy is being challenged by other strategies for the custom development of delivery systems based, for example, on bacteriophage virus like particles (VLPs) [284] or other biomimetic platforms [285]. Future research on exosomes for vaccine purposes will have to establish a comparison with these other types of nano-platforms, particularly in preclinical studies.

3-        While the review effectively synthesizes existing knowledge, the rapidly evolving nature of this field may require updates to incorporate recent research findings for a more comprehensive overview.

We agree that the field is evolving rapidly, but we provide a rather thorough update on the latest research. We have reviewed the most recent literature in the field, in particular on the functions of exosomes in viral infections, their engineering for vaccine purposes and current preclinical trials for antiviral strategies, with particular reference to SARS-CoV-2. We would also like to draw your attention to the fact that we base our work on 289 references, almost a third of which are publications from the last two years (38 of 2022, 55 of 2023, 6 of 2024). However, in line with your request for an update, we have taken advantage of the time between submission at the end of December 2023 and this revision to add a few new references produced in 2024 (see above).

4-        The review could benefit from elaborating on the current status of clinical trials and practical applications.

We would have liked to have had more details about the clinical trials aspect you are requesting. We are aware of the importance of including clinical trials, but unfortunately, none of the preclinical studies concerning exosome-based vaccines to combat viruses have progressed to clinical trials.

As we agree that this transition to clinical trials and practical applications is significant, we have added in the new version of our manuscript, a passage on exosome-based anti-cancer immunotherapies (with 3 new references) that have passed the clinical phase, to initiate a discussion on these aspects.

Line 1313: This strategy has already proven its relevance in oncology, with DC exosome-based vaccines that have passed both preclinical and clinical phases, demonstrating good tolerance and safety [287], [288]. On the downside, the many clinical trials carried out in recent years with exosome-based cancer immunotherapies have not always produced the expected results, prompting further research into the immune mechanisms controlled by exosomes in vivo [289].

Concerning this strategy against viruses:

Line 1244: “In this respect, an extremely promising vaccine candidate that has passed pre-clinical trials proposes the inhalation delivery of pulmonary exosomes, which carry the recombinant SARS-CoV-2 RBD.”

 We also mentioned in line 1274: “However, exosome-based vaccines against viruses are still in their infancy and, so far, they have not produced any widely approved formulations.”

Concerning their applications for example as diagnostic biomarkers, we have pointed an example in line 746:

 “It is noteworthy that exosome-based diagnostic tools have already been trialed in clinical settings, demonstrating promising results in guiding clinicians to establish proper therapeutic interventions, as illustrated in this study on high-grade prostate cancer. Similarly for the clinical trials of exosomes as drug vectors. Finally, with regard to exosome application for drug delivery, clinical trials have not yet taken place.” 

Additionally, we added in line 959

“In conclusion, while exosomes show promise as drug vectors, their clinical application still encounters significant challenges.”

5-        Further exploration of ethical concerns surrounding exosome use in research and medicine is needed.

Thank you for highlighting this point. We dedicated the end of paragraph 7 to developing the ethical concerns related to the use of exosomes in line 1319: Finally, ethical considerations surrounding safety, informed consent, and unpredictable consequences must be addressed. Rigorous testing, transparency in research, and clear patient communication are essential. Balancing innovation with ethical standards is paramount to harness the full potential of exosomes in medicine. The evolution of guidelines to ensure responsible and beneficial applications is strikingly evident. 

6-        Discussing potential future research directions would provide a forward-looking perspective.

We now discuss this point with added references in the section 7 challenges and issues, and indicate that the exosome strategy is challenged by other nano-platform production strategies such as bacteriophages- VLPs

Line 1302: In addition to the advantages afforded by standardized liposomes for the delivery of therapeutics, the "exosome" strategy is being challenged by other strategies for the custom development of delivery systems based, for example, on bacteriophage virus like particles (VLPs) [284] or other biomimetic platforms [285]. Future research on exosomes for vaccine purposes will have to establish a comparison with these other types of nano-platforms, particularly in preclinical studies. 

Round 2

Reviewer 2 Report (New Reviewer)

Comments and Suggestions for Authors

The review article is now significantly improved and highlights important aspects of exosomes against viral infection.

This manuscript is a resubmission of an earlier submission. The following is a list of the peer review reports and author responses from that submission.

Round 1

Reviewer 1 Report

Comments and Suggestions for Authors

The authors made a very thorough review of the topic, but it needs to be improved in a few aspects. Please see below the comments,

Please add an overall schematic diagram.

Abstracts need to be updated as per the new introduction. 

For introduction part-

  1. Clarity and Flow:

Simplify sentences for clarity.

Ensure a smooth transition between ideas.

  1. Thesis Statement:

Explicitly state the objective of the review in a concise thesis statement.

  1. Background Context:

Provide a brief background on cellular communication.

Highlight the essential role of signals in reaching target cells.

  1. Transition to EVs:

Clarify the transition to extracellular vesicles (EVs).

Emphasize the transformative discovery of EVs in immune cell crosstalk.

  1. Importance of EVs:

Clearly articulate the significance of EVs in intercellular communication.

Briefly mention their diverse roles and potential applications.

  1. Recent Advances:

Streamline the discussion of recent advances in EV research.

Highlight the diverse nature of EVs and their cargoes.

  1. Focus on Exosomes:

Explicitly state the decision to focus on exosomes.

Introduce the specific aims of the review regarding exosomes.

  1. Connection to Pathologies:

Clarify the link between exosomes and health/pathological conditions.

Highlight the relevance of exosomes in cancer pathologies.

  1. Infectious Diseases and Viral Exploitation:

Clearly introduce the exploration of exosomes in the context of infectious diseases.

Emphasize the role of exosomes in the organism's response to viral infections.

  1. Antigen Presentation and Vaccine Strategies:

Explicitly state the intention to review exosomes' role in antigen presentation.

Clearly convey the relevance of this information to vaccine strategies.

  1. Conclusion:

Summarize the overarching goals of the review.

Provide a brief outlook on the insights the review aims to contribute.

One overall chart will be helpful to understand the vast review.

For the main part of the review.

1.      Clinical Applications:

Explore and discuss current or potential clinical applications of exosomes, such as their use in diagnostics, targeted therapy, or regenerative medicine.

2.      Technological Advances in Exosome Research:

Highlight recent technological advances that have contributed to our understanding of exosomes, such as advanced imaging techniques, single-vesicle analysis, or high-throughput methodologies.

3.      Exosomes in Drug Delivery:

Expand on the potential of exosomes as drug delivery vehicles, discussing their advantages, challenges, and recent developments in this application.

4.      Immunomodulatory Functions of Exosomes:

Explore in-depth the immunomodulatory functions of exosomes, elucidating how they influence immune responses in various physiological and pathological contexts.

5.      Cross-Talk Between Exosomes and Microbiota:

Investigate the interaction between exosomes and the microbiota, exploring how this cross-talk may impact host physiology and immune responses.

6.      Environmental Influences on Exosome Composition:

Discuss how environmental factors, such as diet or exposure to pollutants, can influence the composition and function of exosomes.

7.      Regulation of Exosome Release:

Explore mechanisms that regulate the release of exosomes, including cellular pathways and signaling cascades that modulate their secretion.

8.      Exosomes in Regulating Stem Cell Fate:

Include a section on the influence of exosomes on stem cell fate determination, emphasizing their role in tissue regeneration and repair.

9.      Ethical Considerations in Exosome Research:

Address ethical considerations related to exosome research, particularly in areas such as privacy concerns, consent, and potential misuse of exosome-based technologies.

Comments on the Quality of English Language

Grammar need to be checked in the manuscript. 

Reviewer 2 Report

Comments and Suggestions for Authors

This review article provides a comprehensive overview of the roles of exosome in antigen presentation, as well as their use in therapeutic targets and vaccine strategies, which is of significant relevance in the field. However, several issues need to be addressed:

1.The manuscript lacks coherence and logical flow in its presentation, as it primarily focuses on listing research findings without offering a cohesive analysis or summary of the content. The author should provide a more structured approach to the narrative, integrating the research findings into a comprehensive analysis and conclusion. Doing so will significantly enhance the overall quality and impact of the article.

2. Certain sections of the article appear to be tangential to the main theme. For instance, the discussion on "Exosome biogenesis" in section 2.1 may warrant consideration for its relevance to the overall focus of the article. It is suggested to carefully evaluate the necessity of including such content in the manuscript.

3. Some titles do not effectively summarize the content of the paragraphs. Advise to add some subheading to categorize and summarize the extensive text, allowing readers to better understand the content.

4. Additionally, it is recommended to incorporate more visual aids such as figures and tables to provide a more intuitive summary of the review's content.

5. Could the authors consider incorporating summary tables and figures to provide a more intuitive overview of the review's content? Currently, many findings are listed together without clear visual representation, and including visual aids could enhance the clarity and impact of the manuscript.

Comments on the Quality of English Language

It is recommended refining the language of the article to ensure clarity and accuracy in expression. Some sentence structures and word choices may need adjustments to enhance the fluency and readability of the article.

Reviewer 3 Report

Comments and Suggestions for Authors

The manuscript „Exosome-mediated Antigen Delivery: Unveiling Novel Strategies in Viral Infection Control“ is a well-structured review about a relevant and timely topic that might be of interest for a wide range of different readers.

Some major and some minor points require attention.

1.      Authors refer to EVs as „therapeutic tools for drug delivery“ in line 54. Further, authors state „exosomes can deliver their contents directly to target cells. Their membrane-based nature allows them to fuse with the plasma membrane“ in lines 274-274. Although it is a much sought-after aim, delivering macromolecules to the cytosol in the form of an approved drug has been challenging and has only been achieved rarely and in very limited ways. More in depth explanations should be added here. Reasons why exosomes are not more often used for delivering macromolecular drugs (e.g. proteins or nucleic acids) to the cytosol should be discussed more specifically, including a comment on what is known about the quantitative efficiency of exosomes in delivering macromolecules to the cytosol, to allow readers a comparison with other delivery-strategies for macromolecules.  

2.      The mechanism of drug delivery should be described more in detail to make it comparable to other tools for drug delivery. More detailed information should be added to the sentence „Their membrane-based nature allows them to fuse with the plasma membrane“ in lines 274-274“, including information on the exact method that was used to generate the data that led to this conclusion and which further molecular details of the process are known.  

3.      Widely varying compositions, as described for endosomes, may pose a challenge to the development of a drug that needs to fulfill standardized criteria. It would be valuable for the readers to briefly discuss whether it is feasible or not to manufacture or synthesise entities that fulfill the most essential functions of endosomes in contrast to the biogenesis of endosomes.

4.      To better link the manuscript’s topic to recent vaccine developments, it would be valuable for the readers to more specifically and in a more direct comparison describe which useful features synthetic carriers (as reviewed for example here PMID: 34705260) are lacking compared to exosomes.

Minor points:

5.      The title of section 4.1 is identical with the title of section 2 and seems not to fit very well to the content of section 4.1. A different, more appropriate title for section 4.1 should be found.

6.      Line 559-560, please check this sentence again and rephrase.

7.      Two titles in the manuscript contain the expression „Extravesicular vesicles“, please check whether this was intended or whether this should have been „extracellular vesicles“. If the original expression was intended, please explain the expression before mentioning it for the first time.

8.      Figures in the PDF were just about legible but very blurry, please double-check the image quality.

Comments on the Quality of English Language

-
